# Risk Assessment and Sources Apportionment of Toxic Metals in Two Commonly Consumed Fishes from a Subtropical Estuarine Wetland System

**DOI:** 10.3390/biology13040260

**Published:** 2024-04-14

**Authors:** Md. Moudud Ahmed, As-Ad Ujjaman Nur, Salma Sultana, Yeasmin N. Jolly, Bilal Ahamad Paray, Takaomi Arai, Jimmy Yu, Mohammad Belal Hossain

**Affiliations:** 1Department of Fisheries and Marine Science, Noakhali Science and Technology University, Noakhali 3814, Bangladesh; 2Atmospheric and Environmental Chemistry Laboratory, Chemistry Division, Atomic Energy Centre, Dhaka 1000, Bangladesh; 3Department of Zoology, College of Science, King Saud University, P.O. Box 2455, Riyadh 11451, Saudi Arabia; bparay@ksu.edu.sa; 4Environmental and Life Sciences Programme, Faculty of Science, Universiti Brunei Darussalam, Jalan Tungku Link, Gadong BE1410, Brunei; takaomi.arai@ubd.edu.bn; 5School of Engineering and Built Environment, Griffith University, Brisbane, QLD 4111, Australia

**Keywords:** toxic metals, estuarine fish, health risk, carcinogenic risk, tropical estuary

## Abstract

**Simple Summary:**

This study analyzes muscle and gill samples from 36 samples of 2 popular edible fishes from a subtropical wetland ecosystem to determine the amounts of harmful heavy metals, such as Pb, Hg, Zn, Cu, and Cr. The metal concentrations in the muscle and gills fluctuate, with zinc having the greatest concentration. Pb and Hg levels are higher than those that are acceptable for human consumption, while Cu, Zn, and Cr levels are within safe bounds. High amounts of Zn, Cu, and Cr may pose health hazards, according to risk indices. The research also highlights sources of heavy metals that are caused by humans and the natural world, highlighting the necessity of continual control and monitoring to guarantee the safety of fish for human consumption.

**Abstract:**

The widespread occurrence of heavy metals in aquatic environments, resulting in their bioaccumulation within aquatic organisms like fish, presents potential hazards to human health. This study investigates the concentrations of five toxic heavy metals (Pb, Hg, Zn, Cu, and Cr) and their potential health implications in two economically important fish species (*Otolithoides pama* and *Labeo bata*) from a subtropical estuarine wetland system (Feni estuary, Bangladesh). Muscle and gill samples from 36 individual fish were analyzed using energy dispersive X-ray fluorescence (EDXRF). The results revealed that the average quantities of heavy metals in both fishes’ muscle followed the declining order of Zn (109.41–119.93 mg/kg) > Cu (45.52–65.43 mg/kg) > Hg (1.25–1.39 mg/kg) > Pb (0.68–1.12 mg/kg) > Cr (0.31–5.82 mg/kg). Furthermore, Zn was found to be present in the highest concentration within the gills of both species. While the levels of Cu, Zn, and Cr in the fish muscle were deemed acceptable for human consumption, the concentrations of Pb and Hg exceeded the permissible limits (>0.5 mg/kg) for human consumption. Different risk indices, including estimated daily intake (EDI), target hazard quotient (THQ), hazard index (HI), and carcinogenic or target risk (TR), revealed mixed and varying degrees of potential threat to human health. According to the EDI values, individuals consuming these fish may face health risks as the levels of Zn, Cu, and Cr in the muscle are either very close to or exceed the maximum tolerable daily intake (MTDI) threshold. Nevertheless, the THQ and HI values suggested that both species remained suitable for human consumption, as indicated by THQ (<1) and HI (<1) values. Carcinogenic risk values for Pb, Cr, and Zn all remained within permissible limits, with TR values falling below the range of (10^−6^ to 10^−4^), except for Zn, which exceeded it (>10^−4^). The correlation matrix and multivariate principal component analysis (PCA) findings revealed that Pb and Cr primarily stemmed from natural geological backgrounds, whereas Zn, Cu, and Hg were attributed to human-induced sources such as agricultural chemicals, silver nanoparticles, antimicrobial substances, and metallic plating. Given the significance of fish as a crucial and nutritious element of a balanced diet, it is essential to maintain consistent monitoring and regulation of the levels and origins of heavy metals found within it.

## 1. Introduction

Long-lasting toxic metals, recognized for their enduring presence in the environment, are significant pollutants capable of inducing detrimental effects such as cytotoxicity, mutagenicity, and carcinogenicity in organisms [1,2,3,4]. These contaminants penetrate aquatic ecosystems through various pathways, encompassing surface runoff, untreated wastewater and sewage discharge, deposition of airborne dust and aerosols, agricultural fertilizer application, electronic waste, and industrial effluents [5,6,7,8,9,10]. Geogenic metals can originate from various geological processes, including volcanic activity, mineral formation, and weathering of rocks. Ultimately, these metals enter the food chain, accumulating in aquatic organisms and eventually in humans, leading to persistent adverse effects on human health [11,12].

Both humans and animals come into contact with toxic metals through various pathways of exposure, such as ingestion or dermal contact [11]. As an example, the toxicity of inorganic arsenic (As) can lead to issues such as abdominal pain, vomiting, and diarrhea. Lead (Pb), deemed a non-essential element, can have detrimental health impacts including liver and kidney damage, disruption of skeletal hematopoietic function, and ultimately, fatalities [12]. Chromium (Cr) plays a significant role in insulin function and lipid metabolism. Excessive intake of chromium may lead to pulmonary disorders along with liver and renal dysfunction [13]. Mercury (Hg) is deemed highly toxic, posing a lethal threat to both humans and other organisms. Ingesting high levels of zinc (Zn) can cause gastrointestinal symptoms such as nausea, vomiting, abdominal cramps, and diarrhea. Additionally, prolonged exposure to elevated levels of zinc can potentially damage the liver and kidneys, leading to impaired liver function and kidney failure [11,14].

Fish, serving as a valuable source of high-quality protein and essential micronutrients, holds significant importance in the human diet, particularly in developing nations such as Bangladesh. Moreover, fish serve as widely recognized bio-indicators for assessing heavy metal contamination in aquatic environments, attributed to their capacity for metal accumulation over time [14]. Consequently, the metal concentrations observed in fish tissues and organs are considered indicative of metal levels in water and their subsequent accumulation within the food chain [13]. Therefore, it is crucial to conduct investigations into the accumulation of potentially harmful heavy metals in key fish species and their various organs [14,15,16,17] to ensure that fish consumption does not become a pathway for the transfer of heavy metals to humans [18,19]. *Otolithoides pama* and *Labeo bata* are two widely distributed keystone species within estuarine habitats in Bangladesh, playing a vital role in ecosystem functioning and stability. They are omnivores and herbivores, respectively, and can tolerate a wide range of salinity. As commercially important fish species, these species support the livelihoods of many communities through fisheries and aquaculture activities, contributing to food security and economic development. Conservation efforts aimed at protecting these species are essential for maintaining the ecological balance of estuarine ecosystems and sustaining the socio-economic well-being of dependent neighboring communities.

Both nationally and internationally, regulatory bodies set guidelines and standards for acceptable levels of heavy metals in food, including fish [14]. In addition to these guideline values, various indices such as estimated daily intake (EDI), target hazard quotient (THQ), hazard index (HI), and cancer risk (CR) have been established and are widely employed in assessing ecological and human health risks associated with heavy metal contamination. Evaluating these values assists in determining compliance with regulatory standards [15]. The estimation of EDI provides insights into the quantity of heavy metals individuals are likely to consume daily through fish consumption. THQ, HI, and CR calculations further evaluate potential health risks associated with this exposure, taking into account factors like contaminant concentration in fish and consumption patterns [18]. Exceeding permissible limits may lead to regulatory actions and advisories aimed at safeguarding public health.

The Feni River estuary in Bangladesh plays an important role in meeting local and national protein demands by providing diversified fish and fishery products. However, the estuary is currently experiencing heavy metal contamination, with multiple contributing factors identified. These include population growth, agricultural practices, discharge of industrial and medical waste, haphazard population settlements, fish farming, washing activities, discharge of poultry waste, recreational pursuits, and improper disposal of untreated domestic effluents [20]. While numerous studies have investigated heavy metal contamination in fish globally [13,14,15,16,17,18,21], as well as in various estuaries in Bangladesh, such as the lower Meghna estuary, upper Meghna estuary, and Karnaphuli estuary [4,22,23,24,25,26,27], there is a lack of data on heavy metal contamination and associated human health risks in edible fishes from the Feni River estuarine system. Previous research by Islam et al. [20] indicated sediment contamination in the estuary by certain metals and suggested further investigation at the organismal level. Thus, this study marks the initial step in assessing the concentrations of selected heavy metals (Pb, Hg, Zn, Cu, and Cr) in the muscle and gill tissues of two commonly consumed fish species, *Otolithoides pama* (Hamilton, 1822) and *Labeo bata* (Hamilton, 1822), from the Feni River estuarine system. This study is important, as it allows for a comparison of the Feni River with other aquatic systems, estuaries in particular, and with other species with similar niches. This study aims to explore the following inquiries: (i) What are the concentrations of toxic metals in commonly consumed fish species in the Feni River estuary and what health risks may arise from the consumption of these fish due to their toxic metal content? And (ii) what are the main sources of toxic metal contamination affecting these fish species?

## 2. Materials and Methods

### 2.1. Study Area

The Feni River estuary is located in the southeastern part of Bangladesh, positioned between 22°46′44″ N latitude and 91°22′42″ E longitude (Figure 1) [20]. Originating from the hill ranges of Tripura district, India, this river travels 116 km, passing through several towns and cities in Bangladesh and India before arriving in the Bay of Bengal [20]. The estuary is utilized for harvesting fish, irrigation, the aquaculture sector, raising cattle, cleaning, relaxation, brick-building, releasing sewage, disposing of household trash, and water-based transportation. The area experiences a predominantly seasonal rainfall pattern, with rains occurring from June to November and a dry period spanning December to May. The average annual rainfall measures 3302 mm, and the annual temperature ranges from a maximum of 34.3 °C to a minimum of 14.4 °C [20]. Flowing into the Bay of Bengal, the estuary exhibits salinity levels ranging from 4.20 to 7.50 ppt, with a mean value of 5.78 ± 1.32 ppt [28]. The study area experiences heavy rainfall and annual flooding, shaping fertile alluvial plains along the Feni River. As a result, most catchment areas are conducive to cultivating both temporary and permanent crops, including rice, red pepper, potato, wheat, beans, bananas, tomato, sunflower, and sugarcane [20].

### 2.2. Fish Species Selection and Sample Collection

Fish species selection for this study involved a focus group discussion with fishermen, retailers, and local residents to ensure economic significance and year-round abundance. Ultimately, *O. pama* and *L. bata* were chosen for investigation due to their native freshwater habitat across Bangladesh, including rivers, streams, floodplains, estuarine, and coastal areas. A total of 36 fish samples (Table 1), 6 from each species at each of the 3 sampling sites (S1–S3), were collected along the coast using a mid-water trawl net in February 2021 (Figure 1). The distance between each sampling site was approximately 3 km. Following collection, the samples were promptly preserved in an ice box containing dry ice cubes and transported to the laboratory at the earliest convenience.

### 2.3. Laboratory Analysis

The weight and length of each collected fish were recorded, and subsequently, the edible portions and gills of each individual were stored separately (Table 1). Following this, the samples were transported to the Atmospheric and Environmental Chemistry Laboratory at the Atomic Energy Centre in Dhaka, Bangladesh. The laboratory analysis procedures were detailed by Hossain et al. [4]. In the laboratory, each fish sample underwent cleaning and rinsing with deionized water. Subsequently, the fish were diced using a stainless steel knife sanitized with acetone and hot distilled water prior to use. For metal analysis, both fish flesh and gills were placed in a beaker and subjected to ashing in a muffle furnace at 300 °C for 3 h. The resulting ash samples were then ground into a powder using a carbide mortar and pestle. The powdered sample was compressed into pellets measuring 2.5 cm in diameter using a hydraulic press pellet maker (Specac Ltd, Orpington, UK) with a pressure of 7 tons. Then, the pellet was placed in EDXRF ( energy-dispersive X-ray fluorescence) system for metal analysis. Irradiation of all samples was conducted according to a time-based program controlled by software provided by the EDXRF system. Standard materials underwent irradiation under identical experimental conditions to establish calibration curves for quantitative elemental determination in the respective samples. The concentrations of various elements in fish samples were determined using energy-dispersive X-ray fluorescence (EDXRF) spectrometry. To ensure quality assurance and control (QA/QC), standard reference materials were prepared and analyzed following the same procedure as employed for the experimental samples. The precision, as indicated by the relative standard deviation of the samples, consistently ranged between 3 and 5%. The analyzed accuracy, determined by the relative error for standard reference materials, was below 5%, and the recovery percentage for standard reference materials was between 94 and 106% (Appendix A). During the analysis, analytical blanks and standard reference materials were run to validate data and confirm the accuracy and precision of the analytical method. The detection limits (DL) for the metals were as follows: Pb = 0.01, Hg = 0.005, Zn = 1.00, Cu = 0.03, and Cr = 0.3. Metal contents were expressed in mg/kg wet weight of fresh fish.

### 2.4. Health Risk Assessment

#### 2.4.1. Target Hazard Quotient

The Target Hazard Quotient (THQ) serves as an indicator of the risk associated with exposure to pollutants [29]. The ratio values below suggest that there are no substantial chronic-toxic risks. The equation utilized to calculate THQ is as follows:THQ = (EF × ED × FIR × CF × CM)/(WAB × ATn × RfD) × 10^−3^

Here, EF represents exposure frequency (365 days/year) [30], ED denotes exposure duration (70 years for non-cancer risk, following USEPA guidelines [29] and Yi et al. [31]. FIR stands for fish ingestion rate (7 g/person/day) [32], CF is the conversion factor (0.2) used to convert fresh weight (Fw) to dry weight (Dw) [33]. CM represents metal concentration in fish (mg/kg Dw), WAB denotes the average body weight (60 kg) [34], ATn represents the average exposure time for non-carcinogens (EF×ED) as utilized in characterizing non-cancer risk, and RfD is the reference dose of the metal (3.0 × 10^−4^ mg/kg/day for Hg, 1.0 × 10^−3^ mg/kg/day for Cd, 4.0 × 10^−3^ mg/kg/day for Pb, 3.0 × 10^−4^ mg/kg/day for As, and 4.0 × 10^−2^ mg/kg/day for Cu) [29].

#### 2.4.2. Hazard Index (HI)

Considering the assumption of cumulative effects arising from a combination of trace elements in fish, the hazard index (HI) was utilized to assess the risk associated with multiple contaminants, using the following approach:HI = Σ THQs
where ‘s’ is the different trace elements [35].

#### 2.4.3. Target Cancer Risk

In the context of carcinogens, the risks are assessed based on the additional probability of an individual developing cancer over their lifetime due to exposure to a potential carcinogen, termed the incremental or excess individual lifetime cancer risk [36]. Acceptable risk levels for carcinogens typically fall within the range of 10^−^^4^ to 10^−^^6^ [37]. The Target Cancer Risk model (TR) is determined by multiplying the oral carcinogenic potency slope of inorganic As by its exposure level, and it is employed to estimate the carcinogenic risk associated with inorganic As over a lifetime [35]. The following formula was used:TR = (EF × ED × FIR × CPSo × CM)/(WAB × ATn) × 10^−^^3^

Here, CPSo stands for the oral carcinogenic potency slope. Among the examined trace metals, arsenic (As) is recognized for its carcinogenic properties [35].

#### 2.4.4. Estimated Daily Intake (EDI)

The daily intake of metals is influenced by both the concentrations of metals in food and the daily food consumption [38]. The estimated daily intake (EDI) is determined through the following equation:EDI = CM × [(DC fish)/BW]

Here, CM represents the concentration of heavy metals in fish muscles (mg/kg). DC denotes the daily fish consumption (g/day) per capita for the Bangladeshi population (7 g/day). Lastly, BW signifies the average body weight of an adult in Bangladesh (60 kg) [32].

#### 2.4.5. Source Identification

In order to identify probable origins of pollution, principal component analysis (PCA) was applied to the heavy metals determined in the gill and muscle tissues of the studied fish species. The loading plot and the scores were used to explain the associations between the variables (heavy metals) and the samples (fish species) [4,20]. The most commonly used multivariate statistical tactic, the correlation matrix (Pearson’s correlation), was also utilized for searching the noteworthy correlations between the heavy metals in the studied samples.

## 3. Results and Discussion

### 3.1. Heavy Metals Concentration in Fish Species

The mean concentrations of Pb, Hg, Zn, Cu, and Cr in the muscle and gill of the studied species (*O. pama* and *L. bata*) from the Feni River estuary are presented in Table 2. The analyzed heavy metals in both fishes’ muscles followed the decreasing order of Zn > Cu > Hg > Pb > Cr. The maximum concentration of Zn, Pb, and Cr was observed in the muscle of *L. bata*, whereas *O. pama* had the highest load of Cu and Hg. In the case of gills, Zn and Cu were found abundantly in both species. In the gills, the highest concentration of Zn was recorded in *L. bata*, while Cu exhibited the highest concentration in *O. pama* (Table 2). The variation in metal concentration among fish could be attributed to their feeding habits, ecological needs, metabolism, and accumulation capacity of each species [39,40]. However, statistically, the average metallic concentrations per species did not vary significantly (*p* > 0.05).

Additionally, the average concentration of metals in the benthic fishes was found to be higher in the previous studies [4,41]. The *O. pama* species shows omnivorous behavior, where they consume a variety of food items including algae, aquatic plants, small invertebrates, and detritus found in their habitat. The fish species *L. bata* typically exhibits herbivorous feeding habits, primarily consuming algae, aquatic plants, and detritus present in its habitat.

Previous studies reported sediment as the primary pathway through which fish uptake heavy metals, as they ingest human substances and benthic invertebrates present in the sediment. [40,42]. Hence, benthopelagic fish may exhibit higher metal load than pelagic fish [41,43], which is also consistent with the present findings, indicating that the presence of metal in fish is not only influenced by the feeding habit but also the habitat they graze on [44,45,46,47].

The results of this study were compared to findings from various studies of tropical Asian regions (Table 3), revealing that the metals analyzed (Pb, Hg, Zn, Cu, and Cr) were also detected in fish muscle and gills from diverse geographic locations. While the concentrations of Zn and Cu in fish muscle were higher compared to the studied fish species from the Red Sea, Egypt [33], Mediterranean Sea, Turkey [48,49], and Gulf of Cambay, India [50], they remained within the standard values outlined by the FAO [51] indicating that despite higher levels in comparison to other regions, they did not exceed internationally recognized safety thresholds. However, the concentrations of Pb and Hg in fish muscle in this study exceeded the established standards (Table 3). In our study, the levels of Pb, Zn, and Cr observed in fish gills were notably lower than those reported in fish sampled from the contaminated Bangshi River [52]. This stark difference can be attributed to the pronounced contamination issues plaguing the Bangshi River, including unregulated discharge from the Dhaka Export Processing Zone (DEPZ) and the adjacency of pharmaceutical industries, poultry farms, and a tannery along its banks. In contrast, the Feni River estuary, while facing its environmental challenges, does not exhibit the same degree of heavy metal pollution as observed in the Bangshi River. Nevertheless, the presence of these metals in this study aligns with previous research on fish muscles [31,33]. Metals such as Pb and Hg are commonly associated with sediments and are vital sources of contamination for benthic fish [53]. Moreover, the probable sources of Hg in the Feni River estuary are linked to industrial discharges and the deposition of atmospheric pollutants from coal-fired brickfields along the river [54]. Conversely, Cu and Cr in the aquatic environment may originate from textiles, dyeing and tanning industries, photography, battery production, paint and ink manufacturing, and runoff from upstream agricultural fields [20,52,55].

### 3.2. Health Risk Assessment

Metal concentrations in fish muscle were utilized to evaluate the potential health risks posed to the local population through fish consumption. Risk indices were computed by comparing these concentrations with the maximum permissible limits for human consumption set by the Food and Agriculture Organization [51]. The results indicated that the levels of Pb and Hg detected in the muscle tissue of the analyzed fish surpassed the recommended threshold of 0.5 mg/kg. In contrast, Zn and Cu concentrations in all samples were below the FAO guidelines (Table 3).

#### 3.2.1. Target Hazard Quotient (THQ) and Hazard Index (HI)

The target hazard quotient (THQ) was computed for each heavy metal present in the fish species under examination, with a recommended threshold of 1 [29]. THQ values below 1 suggest that the exposure level falls beneath the reference dose, indicating unlikely adverse effects over an individual’s lifetime [31]. In this investigation, THQ values ranked in descending order as follows: Hg > Cu > Cr > Zn > Pb, for both fish species (Table 4). Nevertheless, the THQ values for all examined metals at various stations and across all species remained within safe limits for human consumption.

Furthermore, the hazard index (HI) was determined for all metals. An HI exceeding 1 signifies the potential toxicity and health hazards posed by the metals [65,66,67]. The findings of this research revealed that *O. pama* exhibited a higher HI value compared to *L. bata*, yet both were deemed safe for consumption.

#### 3.2.2. Target Lifetime Carcinogenic Risk (TR)

The Target Cancer Risk (TR) from consuming fish was assessed for Pb, Cr, and Zn. The TR for Pb, Zn, and Cr in fish is an important measure used to evaluate the potential health hazards of consuming fish contaminated with these heavy metals. The metric indicates the projected likelihood of an individual acquiring cancer throughout their lifespan due to exposure to these particular pollutants. In this study, the average TR values for Pb, Cr, and Zn from the consumption of *O. pama* were found to be 2.632 × 10^−6^, 8.542 × 10^−4,^ and 1.438 × 10^−3^, respectively (Table 5). Conversely, in the case of *L. bata*, the TR values for Pb, Cr, and Zn were 3.422 × 10^−6^, 4.969 × 10^−4^, and 1.365 × 10^−3^, respectively (Table 5). These results indicate that the estimated carcinogenic risk associated with consuming *L. bata* is lower compared to *O. pama* for the heavy metals analyzed. Although the results show very low risks, they suggest that individuals consuming *O. pama* may have a slightly higher risk of developing cancer over their lifetime due to exposure to Cr, and Zn compared to those consuming *L. bata*. As per established guidelines, a TR value of 1 × 10^−6^ is commonly used as a benchmark for acceptable risk in environmental and public health risk assessments. TR values below this threshold are typically considered to pose an acceptable level of risk and above cause significant risk. For fish, these TR values are typically grouped into three categories: TR < 10^−6^ is considered negligible, 10^−6^ < TR < 10^−4^ falls within an acceptable range, and TR > 10^−4^ is deemed unacceptable [36,37,68]. In this study, carcinogenic risk values for Pb and Cr were found to be within acceptable limits (Table 5). However, for Zn, the risk exceeded the acceptable limit (Table 5). Despite the examined fish species being considered safe for human consumption in the present study, there is a potential risk of developing cancer with continuous consumption of over 70 years. Nevertheless, it is crucial to acknowledge that the definition of “safe” can differ based on the context and the unique conditions of the exposure. Moreover, there may be variations in regulatory norms across different countries or areas. Hence, it is necessary to refer to pertinent standards and laws that are specific to the particular location of concern to ascertain safe TR values for fish intake.

#### 3.2.3. Estimated Daily Intake (EDI) of Heavy Metals

The estimated daily intake (EDI) of heavy metals provides valuable insight into the quantity of these contaminants that individuals are likely consuming basis. This method highlights the levels of nutrients, contaminants, and bioactive compounds ingested, offering valuable insights into potential dietary deficiencies or exposure to food element [69]. The intake data can then be utilized to analyze a specific element of interest. This study assessed the dietary exposure to five trace elements (Pb, Hg, Zn, Cu, and Cr) through the consumption of fish in a regular human diet and measures the dietary intake. Table 6 displays the estimated daily intake (EDI) of heavy metals for *O. pama and L. bata*. The findings suggest that the EDI values for all elements examined were greater in *O. pama* than in *L. bata*, except for Pb. On average, consumers of *O. pama* are projected to have a higher intake of metals such as Hg, Cr, and Zn compared to consumers of *L. bata*. Nevertheless, the consumption of Pb was greater for *L. bata*. The data indicate that there could be differences in the accumulation and distribution of metals between the two fish species. Specifically, *O. pama* may have higher amounts of some metals in their tissues compared to *L. bata*.

To evaluate the potential health risk to consumers, the findings were compared to the maximum tolerable daily intake (MTDI) for heavy metals (Table 6). The guideline value signifies the upper limit of a specific metal that an individual can ingest daily throughout their lifetime without encountering any negative health consequences [38]. It serves as a benchmark for evaluating the safety of consuming food and determining the acceptable quantities of heavy metals in food samples. Regulatory bodies normally determine MTDI values through scientific evaluations of the toxicity and health concerns linked to exposure to particular heavy metals. In this study, the EDI values for Hg and Zn in both fish species surpassed the maximum tolerable daily intake (MTDI) limit. These findings suggest that people who eat these types of fish may be at risk of absorbing levels of mercury and zinc beyond the recommended safe daily intake for a lifetime, which could lead to negative health consequences. The surpassing of the maximum tolerable daily intake levels for Hg and Pb raises concerns over the potential health hazards linked to the intake of fish, which may result in heavy metal exposure. Mercury (Hg) is specifically recognized for its ability to cause damage to the nervous system, while consuming too much Zn can result in digestive problems and other health complications [37]. Hence, these findings emphasize the significance of monitoring the levels of heavy metals in fish and implementing steps to reduce exposure, to guarantee food safety and safeguard public health. Nevertheless, these figures consider variables such as an individual’s body weight, the rate at which a substance is absorbed, and the possible cumulative consequences of prolonged exposure.

#### 3.2.4. Principal Component Analysis (PCA) and Correlation Matrix

Principal component analysis (PCA) serves as a tool to discern patterns, similarities, or variations among different metals based on their characteristics, facilitating classification, clustering, or further examination [70]. In our investigation, two principal components were sufficient to explain the entirety of the variance (100%). The PCA results show that the first component explained 69.79% of the total variation in the dataset, while the second component accounted for an additional 30.21% of the variance (Figure 2). Components with higher percentages of variance capture a greater amount of information regarding the underlying structure of the data. Hence, the first component, characterized by its significant variance percentage, accounts for a substantial proportion of the variability present in the dataset. The loadings of various metals on each component offered valuable insights into the interconnections between variables. In this instance, the first component displayed significant loadings of Zn and Cu, suggesting a robust link between these metals in the dataset. This indicated a strong correlation between the levels of Zn and Cu in the samples. In contrast, the second component showed a strong association with Hg, indicating that changes in Hg levels are less influenced by Zn and Cu levels and may indicate a unique pattern of variability in the dataset. In general, these findings indicate a strong correlation between the levels of Zn and Cu, although the variability in Hg contents differs among the analyzed samples. (Figure 2). Furthermore, the PCA outcomes were corroborated by Pearson’s correlation analyses, revealing strong linear relationships between Zn and Cu (0.82) and Cu and Hg (0.79) at a significance level of 0.05 (Table 7). Such correlations imply that Zn, Cu, and Hg may share common origins, hinting at a reasonable connection among these heavy metals within the environment [71,72,73]. These findings suggest that the presence of heavy metals in fish species likely stems from diverse sources, whether anthropogenic (attributed to human activities) or natural [4,20,71]. Given that the measured levels of these metals in the study area exceeded standard values, they probably originated from anthropogenic activities within the river’s catchment area, spanning from upstream to downstream. This catchment area is predominantly utilized for agricultural activities, human habitation, aquaculture, fishing, metal galvanization, and certain pharmaceutical industries [20]). Consequently, the widespread application of agrochemicals in croplands, discharge of household and industrial wastes, operations of upstream metal plating facilities, and the presence of airborne particles could potentially account for the elevated concentrations of Zn, Cu, and Hg in the area [20]).

## 4. Conclusions

This study aimed to assess the contamination levels and associated health risks posed by five heavy metals (Pb, Hg, Zn, Cu, and Cr) in two commercially significant fish species from the Feni River estuary. Analysis of heavy metal concentrations in the muscle tissue of both fish species revealed a descending order of Zn > Cu > Hg > Pb > Cr. The highest concentrations of Zn, Pb, and Cr were observed in the muscle tissue of *Labeo bata*, whereas *Otolithoides pama* exhibited the highest levels of Cu and Hg. The levels of studied metals in the gills of the two species varied. Disparities in heavy metal presence in the fish species suggest that dietary habits and habitat may influence metal accumulation. While levels of Pb and Hg exceeded permissible thresholds for human consumption, assessments of target hazard quotient (THQ), hazard index (HI), and estimated daily intake (EDI) indicated that consuming fish from the studied region did not pose significant health risks, except for potential cancer hazards for Zn. The notable correlations among Zn, Cu, and Hg imply potentially shared origins, whether anthropogenic or natural. Although isotopic analyses could provide definitive evidence of metal origins, possible sources may include widespread use of agrochemicals in croplands, household waste, upstream metal plating activities, and other human activities within the catchment area. Given that fish continues to be a vital and healthy component of a balanced diet, it is essential to recognize the potential human health risks of contaminated fish. Therefore, it is advisable to maintain regular monitoring of toxic metals in riverine fish.

## Figures and Tables

**Figure 1 biology-13-00260-f001:**
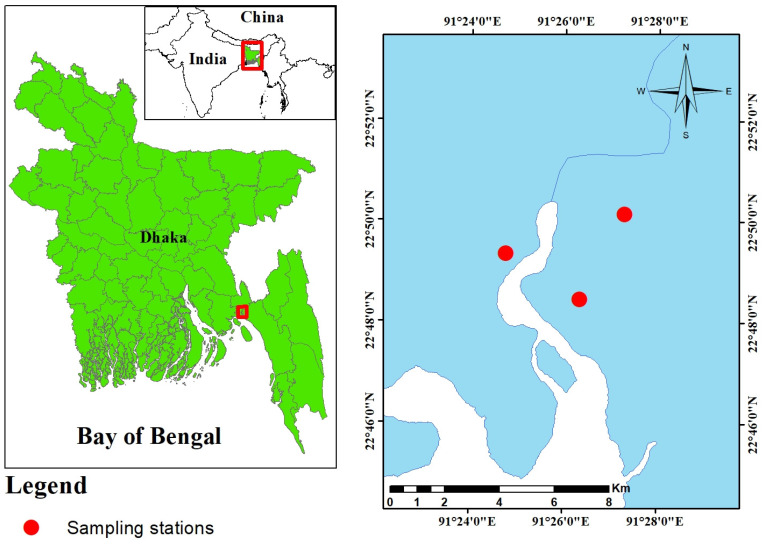
Map of the study area and location of the sampling sites (red circle) in the Feni River Estuary.

**Figure 2 biology-13-00260-f002:**
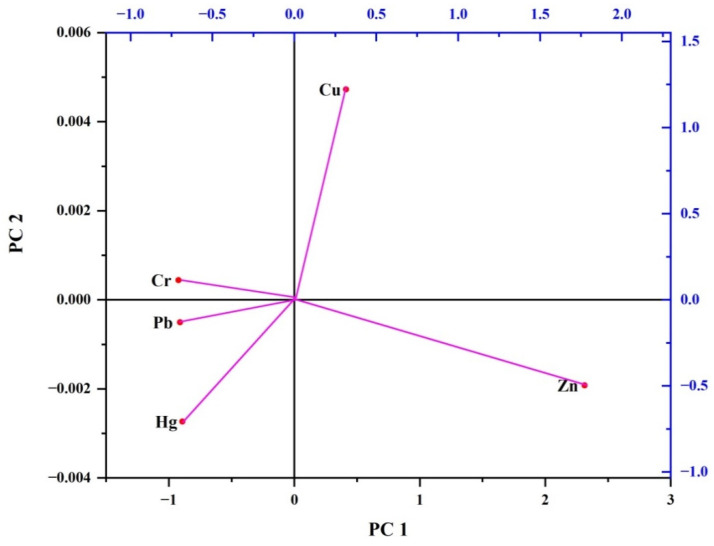
Principal component analysis (PCA) biplot of log-transformed metal concentration in fish tissue.

**Table 1 biology-13-00260-t001:** Biometrics data (mean ± SD) of the selected fish samples from the Feni River.

Sample	Local Name	N	Preference of Feed	Habitant	Body Weight (g)	Length (cm)	Gill Weight (g)
*O. pama*	Poa	18	Small invertebrates and herbivore (on algae)	Pelagic	87 ± 11	22.5 ± 4	2.98 ± 0.8
*L. bata*	Bata	18	Filter Feeders (benthic invertebrates)	Benthopelagic	61 ± 8	18.8 ± 3.5	2.32 ± 06

**Table 2 biology-13-00260-t002:** Summary of the heavy metal concentration in the muscle and gill of the studied fish species (mg/kg).

Species	Elements	Organ	Mean Value + SD	Minimum	Maximum
*O. pama*	Pb	Muscle	0.772 ± 0.083	0.689	0.857
		Gill	0.658 ± 0.102	0.557	0.762
	Hg	Muscle	1.945 ± 0.036	1.906	1.978
		Gill	2.522 ± 0.172	2.412	2.720
	Zn	Muscle	110. 85 ± 1.236	109.411	111.586
		Gill	110.158 ± 1.236	128.461	155.296
	Cu	Muscle	59.821 ± 5.839	53.783	65.438
		Gill	59.821 ± 5.839	45.645	59.212
	Cr	Muscle	<DL ^a^	<DL ^a^	<DL ^a^
		Gill	3.678 ± 1.859	2.595	5.825
*L. bata*	Pb	Muscle	0.974 ± 0.139	0.846	1.123
		Gill	0.884 ± 0.092	0.784	0.967
	Hg	Muscle	1.326 ± 0.066	1.258	1.390
		Gill	1.360 ± 0.097	1.282	1.469
	Zn	Muscle	119.382 ± 0.736	118.544	119.928
		Gill	122.359 ± 9.969	110.853	128.409
	Cu	Muscle	50.088 ± 4.085	45.526	53.407
		Gill	50.088 ± 4.085	47.864	57.914
	Cr	Muscle	3.862 ± 0.786	3.134	4.696
		Gill	<DL ^a^	<DL ^a^	<DL ^a^

^a^ Values were below the detection limits.

**Table 3 biology-13-00260-t003:** Comparison of heavy metals concentrations in this study with other studies and FAO standard values (mg/kg). The combined metal values are derived from both species and three stations.

Area	Pb	Hg	Zn	Cu	Cr	Reference
Muscle						
Feni River Estuary (Bangladesh)	0.68–1.12	1.25–1.39	109.41–119.93	45.52–65.43	0.31–5.82	Present study
Musa estuary (Persian Gulf)	0.07–0.77	0.56–14.00	NA	1.37–3.14	NA	[56]
Red Sea (Egypt, Jordan)	0.21–0.88	NA ^a^	1.9–35	0.22–0.63	1.0–10.3	[33]
Mediterranean Sea (Turkey)	2.98–5.57	NA	3.51–53.5	2.19–4.41	0.07–1.48	[48,49]
Black Sea (Turkey)	0.68	NA ^a^	NA ^a^	1.55	NA ^a^	[57]
Yangtze River (China)	0.117	0.043	NA	1.020	NA ^a^	[31]
Bushehr (Persian Gulf)	0.68	0.86	NA ^a^	NA ^a^	NA ^a^	[58]
Gulf of Cambaya (India)	1.09	NA ^a^	38.24	2.37	0.77	[50]
Hendijan (Persian Gulf)	NA	0.13	NA ^a^	NA ^a^	NA ^a^	[59]
Standard value ^d^	0.5	0.5	1000	70	NA ^a^	[51]
Gill						
Feni River Estuary (Bangladesh)	0.55–0.96	1.28–2.72	110.85–155.29	45.64–59.21	<DL ^c^	Present study
Taihu Lake fish samples	0.49	NA ^a^	NA ^a^	0.24	0.16	[60]
Pulicat Lake, India	1.1	NA ^a^	ND ^b^	1.3	0.2	[61]
ROPME	0.01–1.28	1	NA	0.05–19.5	2.3	[62]
Tigris River in Baghdad	1.50	ND ^b^	1.05	1.10	2.20	[63]
Bangshi River	7.36	0.39	183.64	41.19	4.36	[52]
Langkawi Island	1.00	1.47	49.39	11.55	NA ^a^	[64]

^a^ Not available; ^b^ not detected; ^c^ values were below the limits of detection; ^d^ Food and Agricultural Organization.

**Table 4 biology-13-00260-t004:** Target hazard quotient (THQ) for different heavy metals and their hazard index (HI).

Species	Station	Pb	Hg	Zn	Cu	Cr	HI
*O. pama*	Station-1	0.015	0.632	0.034	0.112	0.099	0.892
	Station-2	0.014	0.672	0.037	0.130	0.074	0.927
	Station-3	0.016	0.624	0.033	0.120	0.168	0.961
	Average of THQ	0.015	0.642	0.035	0.121	0.114	0.930
*L. bata*	Station-1	0.022	0.378	0.035	0.107	0.095	0.637
	Station-2	0.018	0.365	0.032	0.101	0.135	0.651
	Station-3	0.019	0.411	0.034	0.120	0.090	0.674
	Average of THQ	0.019	0.384	0.033	0.109	0.107	0.646

**Table 5 biology-13-00260-t005:** Target risk (TR) values of different species in different stations.

Species	Station	Pb	Cr	Zn
*O. pama*	Station-1	2.603 × 10^−6^	7.443 × 10^−4^	1.766 × 10^−4^
	Station-2	2.474 × 10^−6^	5.605 × 10^−4^	2.007 × 10^−3^
	Station-3	2.821 × 10^−6^	1.258 × 10^−3^	2.031 × 10^−3^
	Average of TR	2.632 × 10^−6^	8.542 × 10^−4^	1.438 × 10^−3^
*L. bata*	Station-1	3.848 × 10^−6^	7.124 × 10^−4^	1.226 × 10^−3^
	Station-2	3.218 × 10^−6^	1.014 × 10^−4^	1.296 × 10^−3^
	Station-3	3.200 × 10^−6^	6.769 × 10^−4^	1.575 × 10^−3^
	Average of TR	3.422 × 10^−6^	4.969 × 10^−4^	1.365 × 10^−3^

**Table 6 biology-13-00260-t006:** Estimated daily intake (EDI) from fish consumption by local residents (mg/kg/day).

Location	Fish Species	Pb	Hg	Zn	Cu	Cr
Station 1	*O. pama*	0.306	0.951	0.104	22.60	1.493
	*L. bata*	0.452	0.570	0.722	21.56	1.431
Station 2	*O. pama*	0.291	1.011	1.181	25.897	1.124
	*L. bata*	0.378	0.550	0.763	20.234	2.034
Station 3	*O. pama*	0.331	0.940	1.196	24.067	2.535
	*L. bata*	0.376	0.619	0.927	24.119	1.358
${MTDI\;}^{a}$		0.40	0.5–1.0	0.90	3–30	0.5–2.0

^a^ Maximum tolerable daily intake.

**Table 7 biology-13-00260-t007:** Pearson correlation analysis of heavy metals in fish tissue.

	Pb	Cr	Cu	Hg	Zn
Pb	1				
Cr	−0.39	1			
Cu	0.50	−0.72	1		
Hg	−0.75	−0.22	0.79 **	1	
Zn	0.26	−0.940	0.82 **	−0.69	1

(*p* < 0.05 **).

## Data Availability

Data are provided in the article.

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
