# Peer review of "Risk Assessment and Sources Apportionment of Toxic Metals in Two Commonly Consumed Fishes from a Subtropical Estuarine Wetland System"

_biology, 2024, doi:10.3390/biology13040260_

Round 1

Reviewer 1 Report

Comments and Suggestions for Authors

In general, the abstracts need to be re-written. The writing is confusing and not informative at all, assuming that the reader knows the indexes and legal limits. Furthermore they highlight "conclusions" that are not supported by this study's results.

In detail:

L38 "above the permissible limits" Adding the values of Pb and Hg limits will make it more informative.

L41 "consumers of these fish may experience health issues " Please re-write. Sounds speculative or as if the study includes effects on human health.

L42 "MTDI" What's that? Better say what it is right away. The reader is not obliged to know.

L43 "Both species were shown to still be suitable for consumption by human beings as per the THQ ..." Please re-write to make it more informative. Add info about THQ and HI limits

 L44 "range for TR or carcinogenic risk values" It's confusing. Is this one or more indexes? You'd better explain shortly that these indexes are calculated for each metal. The reader is not obliged to know. 

L49 "significance" write importance instead

The introduction could be improved, by including some more info about the studied fishes and previous results in estuaries. There should be a short paragraph about the hazard indexes that are calculated, what they're based on and what's their meaning.

In detail:

L56  "infiltrate". Please replace by a another term.

L58-59 The geogenic origin seems to be forgotten in this list, although it can be very relevant in most environments. Not to be ignored.

L60 "infiltrate". Please replace by a another term.

L60-61 "in humans, leading to persistent adverse effects on biogeochemical cycles within the environment" This sentence is very confusing. As it is written, it's hard to recognize its relevance. You focused on humans and suddenly jump back to biogeochemical cycle...confusing...

L66-67 "The metal concentrations found in fish tissues and organs reflect the levels of metals in water..." To a certain extent, right? Depends on the species and the analysed organ. Re-write as to tone down such certainity. 

L70 "pathway" Please use another word.

L73 "crucial for meeting local and national protein demands through fish and fishery products," Rewrite please

L75 "This contamination is attributed..." Say increase in contamination? And again never exclude geogenic contamination....

L75-78 Are there any references, any reports, stats about the activities on Feni River basin? If yes, please cite them.

L80-81 "...no study has specifically addressed heavy metal contamination in edible fishes from the Feni River estuarine wetland system." Instead of using this sentence to highlight the relevance of this study, that is arguing that there are no studies on fish contamination by heavy metals in Feni River, I would add references to studies on fish heavy metal contamination in estuaries + studies refering to the same or close species.  This study is important as it allows comparing Feni River with other aquatic systems, estuaries in particular and with other species, with similar niches. So the relevance of the study should not be reduced to a regional interest in the introduction.

L83 " fish species (Otolithoides pama and Labeo bata)"Maybe you could make a short mention to its habitat (pelagic or benthis), feeding habits, etc...

L85 "within the subtropical estuarine wetland system" Here I would write Feni river instead of "subtropical....system"? or a subtropical estuary with XXX characteristics? You may and should compare with other estuarine systems, but you will not be able to generalize the results fo Feni River to any estuarine wetland system... 

L86-87 "(ii) What health risks may arise from the consumption of these fish due to their toxic metal content"  Written as it is, it sounds speculative. You're not studying the effects on humans. You just calculating indexes. Better fuse with the previous question (i).

The methods present serious flaws. My main concern is the experimental design and the interpretation of results, which do not support the conclusions. I believe that the number of individuals sampled per species and per site is too low. If the authors can consider each sampling place as replicates, then it will be acceptable if assumed to be a preliminary study or screenning. Otherwise, I don't think it can be published. Even if the number of samples individuals was higher the used stats are hardly enough to make the interpretation of these data. A PCA is adequate, but not made this way. 

In detail: L99 "the annual temperature" is that air temperature? What about water temperature?

L102-105 Summarize

L108-112 Summarize

L133 "morpho characteristics" Delete

L137 "laboratory analysis procedures were detailed by Hossain et al." Neverthless you should summarize them here, with the necessary adaptations. 

L138 " in sediment samples" ???? I'm missing the description for sediment collection. Furthermore, there are no results on sediment samples in the Results section. 

L153 "The Target hazard quotient" There should be some more information about this index here or in the introduction

L157-166 The explanation is not clear at all. Please re-write.

L161 " CM represents fish heavy metal concentration" For those who don't know the index this is not clear at all. To start with not saying the the quotient is calculated for each metal, not in the beggining nor in its parcels.

L179 "10-4 to 10-6" -4 and -6 should be superscript

L196  "identify probable origins of pollution" What do you mean by origin? Species? Place?

Results and Discussion. In general, all the considerations about the possible sources of heavy metal fish contamination should be avoided in the Conclusions and Abstract sections as there is no data on the anthropogenic impact around the sampling sites and so they're not considered in the statistics. There is no discussion on the "differences" (not tested) between the two studied species. 

In detail:

L209 Labeo bata should be italic

L211-213 Re-write the whole sentence, please. It's very confusing.

L214 Delete "However"

L214-215 The differences between O.pama and L.bata should have been tested and discussed.

Table 2 I can see the exact same values for Zn and Cu in muscle and gill. Please check this or explain.

L221-222 There should be also a sentence with mention to O.pama and L.bata habitat and feeding habits.

L221-226 This discussion is relevant but it is cleary missing the comparison with the results herein presented with reference to  O.pama and L.bata habitat and feeding habits. You should be discussing this paper results, as compared to all those. The way it is written, it is just pointless. 

L231 "in fish muscle" is this from all fish from both species and the 3 sites? It's not clear. Should be clarified and the approach should be explained.

L235-236 "were lower than those found in fish from the Bangshi River" You could discuss this comparison with Bangshi River a little more. Any data about this river? Compare the number of inhabitants, houses, activities in its water basin, with that Feni's basin?

L237 " on fish organs" which organs? Bibliographic references needed.

L238  "and serve as long-term sources of contamination..." Please re-write

L239-240 "Additionally, the presence of Hg in this study could be attributed to industrial discharges" That's a bit of a stretch. Your results do not support that conclusion. You can mention the possible Hg sources in Feni basin, without saying that the Hg levels found in this study are due to.... For example, the next sentence (on Cu and Cr) is just fine. 

Table 3 Do these values refer to all fish together, 2 species and 3 sites? If so, it should be clearly stated on the table's legend. refers to the Standard values in the table?; because d never shows up in the table

L247-253 Why not fuse this paragraph with 3.1?  The point of measuring heavy metal concentrations was assessing health risk coming from these species consumption, right?

L247 "To assess the health risk to the local population from fish consumption, metal " Re-write please.

L263-264  "where an HI surpassing 1 raises concerns regarding health risks to the local populace"  The next sentence says the same, but more clearly.

Table 4 Highlight HI, by putting it in bold, for example...

L272 Re-write the whole sentence please.

L273 "and Zn from the consumption of Otolithoides pama" Re-write please

L278 Delete "However"

L279 "while for Zn, it exceeded the limit" Re-write please.

L285 " EDI of heavy metals offers insight into the amount of these contaminants’ individuals are likely ingesting on a daily basis" Re-write please.

L286-288 "By emphasizing the levels of nutrients, contaminants and bioactive compounds consumed, this method provides significant understanding into possible dietary deficiencies or exposition to food allergens" Re-write please.

L294 "can potentially lead to health issues for consumers " Be cautios there. Your results do not support this conclusion. Just comparing with MTDI is enough. 

L295-296 "However, the quantity of fish consumed as well as the concentration of a particular metal in fish determines the quantity of that element can be identified in fish." Un-readable sentence.

L304 "PCA can be beneficial in..." Re-write please.

L306 enough instead of "able"

L309-311 " These results indicated that the presence of heavy metals in fish species was likely entering from various sources, which could be either anthropogenic (resulting from human activities) or natural" I don't understand how you can inferr this from your PCA analysis

L312-314"The wide-spread anthropogenic inputs from inundation croplands, pharmaceutical waste, household waste, upstream metal plating entities and airborne particles were visibly responsible for the high concentrations of Zn, Cu, and Hg" I don't see how you results support this conclusion..

Table 7: Legend, "Shrimp"?; Are all the correlations significant? Not clear....

Conclusions section: some of the conclusions are not supported by the results, which one of these study's major flaws.

In detail:

L337 ", marking the first such study" Delete or re-phrase.

L338 "that Labeo bata exhibited lower levels of heavy metal contamination compared to Otolithoides pama" Not clearly stated or discussed in the previous section.

L345 " risks apart from potential cancer risks" Re-write please.

L345-348 "Correlation and principal component analyses shed light on the potential sources of heavy metals in the environment, which included anthropogenic activities like the use of agricultural chemicals, silver nanoparticles, antimicrobial agents, and metallic plating industries along the estuary" Honestly, I don't see how this can inferred from the presented PCA analysis.

L348-351 Re-write please.

Comments on the Quality of English Language

Overall the english is fine, except for some terms that in my opinion should be replaced. 

Author Response

In general, the abstracts need to be re-written. The writing is confusing and not informative at all, assuming that the reader knows the indexes and legal limits. Furthermore they highlight "conclusions" that are not supported by this study's results.

Response: Thank you very much for your valuable time and suggestions to improve the MS. We have carefully gone through all of your comments and addressed properly as below. We are grateful to you for your sincerity and valuable suggestions.

The abstract is re-written as

“The widespread occurrence of heavy metals in aquatic environments, resulting in their bioaccumulation within aquatic organisms like fish, presents potential hazards to human health. This study investigates the concentrations of five toxic heavy metals (Pb, Hg, Zn, Cu, and Cr) and their potential health implications in two economically significant fish species (Otolithoides pama and Labeo bata) from a subtropical estuarine wetland system (Feni estuary, Bangladesh). Muscle and gill samples from 36 individual fish were analyzed using Energy Dispersive X-ray Fluorescence (EDXRF). The results revealed that the average quantities of heavy metals in both fish muscle followed the declining order of Zn (109.41-119.93 mg.kg) > Cu (45.52-65.43 mg/kg) > Hg (1.25-1.39 mg/kg) > Pb (0.68-1.12 mg/kg) > Cr (0.31-5.82 mg/kg). In addition, Zn was also detected in the highest concentration within the gills of both species.   The fish muscle exhibited acceptable levels of Cu, Zn, and Cr. However, Pb and Hg surpassed the permissible limits (>0.5 mg/kg) for human consumption.  Different risk indices, including estimated daily intake (EDI), target hazard quotient (THQ), hazard index (HI), and carcinogenic or target risk (TR), revealed mixed and varying degrees of potential threat to human health. According to the EDI values, individuals consuming these fish may face health risks as the levels of Zn, Cu, and Cr in the muscle are either very close to or exceed the maximum tolerable daily intake (MTDI) threshold.  Nevertheless, the THQ and HI values suggested that both species remained suitable for human consumption, as indicated by THQ (<1) and HI (<1) values. Carcinogenic risk values for Pb, Cr, and Zn all remained within permissible limits, with TR values falling below the range of (10-6 to 10-4), except for Zn, which exceeded it (>10-4). The correlation matrix and multivariate principal component analysis (PCA) findings revealed that Pb and Cr primarily stemmed from natural geological backgrounds, whereas Zn, Cu, and Hg were attributed to human-induced sources such as agricultural chemicals, silver nanoparticles, antimicrobial substances, and metallic plating. Given the significance of fish as a crucial and nutritious element of a balanced diet, it is essential to maintain consistent monitoring and regulation of the levels and origins of heavy metals found within it.”

L38 "above the permissible limits" Adding the values of Pb and Hg limits will make it more informative.

Response: Thank you. We added  the values.

L41 "consumers of these fish may experience health issues " Please re-write. Sounds speculative or as if the study includes effects on human health.

Response: Thank you. Re-written.

L42 "MTDI" What's that? Better say what it is right away. The reader is not obliged to know.

Response:  Thank you. We agree with you. Elaborated in this version as MTDI (maximum tolerable daily intake) threshold.”

L43 "Both species were shown to still be suitable for consumption by human beings as per the THQ ..." Please re-write to make it more informative. Add info about THQ and HI limits

Response: Thank you very much. We have re-written as

‘Nevertheless, the THQ and HI values suggested that both species remained suitable for human consumption, as indicated by THQ (<1) and HI (<1) values. Carcinogenic risk values for Pb, Cr, and Zn all remained within permissible limits, with TR values falling below the range of (10-6 to 10-4), except for Zn, which exceeded it (>10-4).’

L44 "range for TR or carcinogenic risk values" It's confusing. Is this one or more indexes? You'd better explain shortly that these indexes are calculated for each metal. The reader is not obliged to know. 

Response: Thank you. It’s the same: Target Cancer Risk (TR). In literature sometimes used TR and sometimes cancer risk are used.

L49 "significance" write importance instead

Response: We corrected it as importance.

The introduction could be improved, by including some more info about the studied fishes and previous results in estuaries. There should be a short paragraph about the hazard indexes that are calculated, what they're based on and what's their meaning.

Response: Thank you very much. We have included some more info about the studied fishes, previous results in estuaries and a short paragraph about the hazard indexes that are calculated.  We modified the introduction as below:  

Long-lasting toxic metals, recognized for their enduring presence in the environment, are significant pollutants capable of inducing detrimental effects such as cytotoxicity, mutagenicity, and carcinogenicity in organisms [1,2,3,4]. These contaminants penetrate aquatic ecosystems through various pathways, encompassing surface runoff, untreated wastewater and sewage discharge, deposition of airborne dust and aerosols, agricultural fertilizer application, electronic waste, and industrial effluents [5,6,7,8,9,10]. Ultimately, these metals enter into the food chain, accumulating in aquatic organisms and eventually in humans, leading to persistent adverse effects on human health [11,12].

Both humans and animals come into contact with toxic metals through various pathways of exposure, such as ingestion or dermal contact [11]. As an example, the toxicity of in-organic arsenic (As) can lead to issues such as abdominal pain, vomiting, and diarrhea. Lead (Pb), deemed a non-essential element, can have detrimental health impacts including liver and kidney damage, disruption of skeletal hematopoietic function, and ultimately, fatalities [12]. Chromium (Cr) plays a significant role in insulin function and lipid metabolism. Excessive intake of chromium may lead to pulmonary disorders along with liver and renal dysfunction [13]. Mercury (Hg) is deemed highly toxic, posing a lethal threat to both humans and other organisms. Ingesting high levels of zinc (Zn) can cause gastrointestinal symptoms such as nausea, vomiting, abdominal cramps, and diarrhea. Additionally, Prolonged exposure to elevated levels of zinc can potentially damage the liver and kidneys, leading to impaired liver function and kidney failure [11,14].

Fish, serving as a valuable source of high-quality protein and essential micronutrients, holds significant importance in the human diet, particularly in developing nations such as Bangladesh. Furthermore, fish are widely acknowledged as bio-indicators for assessing heavy metal contamination in aquatic environments. The metal concentrations found in fish tissues and organs reflect the levels of metals in water and their accumulation within the food chain [13]. Therefore, it is crucial to conduct investigations into the accumulation of potentially harmful heavy metals in key fish species and their various organs [14,15,16,17] to ensure that fish consumption does not become a pathway for the transfer of heavy metals to humans [18,19]. Otolithoides pama and Labeo bata are two widely distributed keystone species within estuarine habitats in Bangladesh, playing a vital role in ecosystem functioning and stability. As a commercially important fish species, these species support the livelihoods of many communities through fisheries and aquaculture activities, contributing to food security and economic development. Conservation efforts aimed at protecting of these species are essential for maintaining the ecological balance of estuarine ecosystems and sustaining the socio-economic well-being of dependent neighboring communities.

Regulatory bodies establish guidelines and standards for permissible levels of heavy metals in food, including fish. Assessing estimated daily intake (EDI), target hazard quotient (THQ), hazard index (HI) and cancer risk (CR) aids in evaluating compliance with these regulations. EDI estimation provides insight into the amount of heavy metals that individuals are likely to ingest daily through fish consumption. THQ, HI, and CR calculations further assess the potential health risks associated with this exposure, considering factors such as the concentration of contaminants in fish and consumption patterns. Exceeding allowable limits can result in regulatory actions and advisories to protect public health.

The Feni River estuary in Bangladesh plays an important role to accomplish for meeting local and national protein demands through by providing diversified fish and fishery products. However, the estuary is currently experiencing heavy metal contamination, with multiple contributing factors identified. These include population growth, agricultural practices, discharge of industrial and medical waste, haphazard population settlements, fish farming, washing activities, discharge of poultry waste, recreational pursuits, and improper disposal of untreated domestic effluents [20]. While numerous studies have investigated heavy metal contamination in fish globally [13, 14, 15, 16, 17, 18, 21], as well as in various estuaries in Bangladesh such as the lower Meghna estuary, upper Meghna estuary, and Karnaphuli estuary [4, 22, 23, 24, 25, 26, 27], there is a lack of data on heavy metal contamination and associated human health risks in edible fishes from the Feni River estuarine system. Previous research by Islam et al. [20] indicated sediment contamination in the estuary by certain metals and suggested further investigation at the organismal level. Thus, this study marks the initial step in assessing the concentrations of selected heavy metals (Pb, Hg, Zn, Cu, and Cr) in the muscle and gill tissues of two commonly consumed fish species (Otolithoides pama and Labeo bata) from the Feni River estuarine system. This study is important as it allows comparing Feni River with other aquatic systems, estuaries in particular and with other species, with similar niches. The study aims to explore the following inquiries: (i) what are the concentrations of toxic metals in commonly consumed fish species in the Feni river estuary and what health risks may arise from the consumption of these fish due to their toxic metal content? and (ii)What are the main sources of toxic metal contamination affecting these fish species?

L56  "infiltrate". Please replace by a another term.

Response: We have replaced the word “infiltrate” by “penetrate”. Thank you.

L58-59 The geogenic origin seems to be forgotten in this list, although it can be very relevant in most environments. Not to be ignored.

Response: Thank you. We agree with you and modified the section adding ‘ Geogenic metals can  originate from various geological processes, including volcanic activity, mineral formation, and weathering of rocks’

L60 "infiltrate". Please replace by a another term.

Response: We have replaced the word “infiltrate” by “enter”. Thank you.

L60-61 "in humans, leading to persistent adverse effects on biogeochemical cycles within the environment" This sentence is very confusing. As it is written, it's hard to recognize its relevance. You focused on humans and suddenly jump back to biogeochemical cycle...confusing...

Response: Thank you. Re-written as “Ultimately, these metals enter into the food chain, accumulating in aquatic organisms and eventually in humans, leading to persistent adverse effects on human health”

L66-67 "The metal concentrations found in fish tissues and organs reflect the levels of metals in water..." To a certain extent, right? Depends on the species and the analysed organ. Re-write as to tone down such certainity. 

Response: Thank you very much. We agree with you and re-written this part as :

Moreover, fish serve as widely recognized bio-indicators for assessing heavy metal contamination in aquatic environments, attributed to their capacity for metal accumulation over time. Consequently, the metal concentrations observed in fish tissues and organs are considered indicative of metal levels in water and their subsequent accumulation within the food chain

L70 "pathway" Please use another word.

Response: We have replaced the word “pathway” by “way”. Thank you.

L73 "crucial for meeting local and national protein demands through fish and fishery products," Rewrite please

Response: The sentence was rewritten as “The Feni River estuary in Bangladesh plays an important role to accomplish national protein demands by providing diversified fish and fishery products. Now a days, this significant estuary is facing heavy metal contamination.”

L75 "This contamination is attributed..." Say increase in contamination? And again never exclude geogenic contamination....

Response: We have re-written this part as below:

However, the estuary is currently experiencing heavy metal contamination, with multiple contributing factors identified. These include population growth, agricultural practices, discharge of industrial and medical waste, haphazard population settlements, fish farming, washing activities, discharge of poultry waste, recreational pursuits, and im-proper disposal of untreated domestic effluents [20].

L75-78 Are there any references, any reports, stats about the activities on Feni River basin? If yes, please cite them.

Response: Yes- there is only one published article regarding heavy metal pollution in sediment of  Feni River basin and we cited the paper here. The reference is given follow:

  1. Islam, M.S.; Hossain, M.B.; Matin, A.; Sarker, M.S.I. Assessment of heavy metal pollution, distribution and source apportionment in the sediment from Feni River estuary, Bangladesh. Chemos. 2018, 202, 25-32.

L80-81 "...no study has specifically addressed heavy metal contamination in edible fishes from the Feni River estuarine wetland system." Instead of using this sentence to highlight the relevance of this study, that is arguing that there are no studies on fish contamination by heavy metals in Feni River, I would add references to studies on fish heavy metal contamination in estuaries + studies refering to the same or close species.  This study is important as it allows comparing Feni River with other aquatic systems, estuaries in particular and with other species, with similar niches. So the relevance of the study should not be reduced to a regional interest in the introduction.

Response: Thank you for your valuable input. We have re-written ( added all the relevant references)  this section as below:

 The Feni River estuary in Bangladesh plays an important role to accomplish for meeting local and national protein demands through by providing diversified fish and fishery products. However, the estuary is currently experiencing heavy metal contamination, with multiple contributing factors identified. These include population growth, agricultural practices, discharge of industrial and medical waste, haphazard population settlements, fish farming, washing activities, discharge of poultry waste, recreational pursuits, and improper disposal of untreated domestic effluents [20]. While numerous studies have investigated heavy metal contamination in fish globally [13, 14, 15, 16, 17, 18, 21], as well as in various estuaries in Bangladesh such as the lower Meghna estuary, upper Meghna estuary, and Karnaphuli estuary [4, 22, 23, 24, 25, 26, 27], there is a lack of data on heavy metal contamination and associated human health risks in edible fishes from the Feni River estuarine system. Previous research by Islam et al. [20] indicated sediment contamination in the estuary by certain metals and suggested further investigation at the organismal level. Thus, this study marks the initial step in assessing the concentrations of selected heavy metals (Pb, Hg, Zn, Cu, and Cr) in the muscle and gill tissues of two commonly consumed fish species (Otolithoides pama and Labeo bata) from the Feni River estuarine system. This study is important as it allows comparing Feni River with other aquatic systems, estuaries in particular and with other species, with similar niches. The study aims to explore the following inquiries: (i) what are the concentrations of toxic metals in commonly consumed fish species in the Feni river estuary and what health risks may arise from the consumption of these fish due to their toxic metal content? and (ii)What are the main sources of toxic metal contamination affecting these fish species?

Kibria, G,, Hossain, M,M,, Mallick, D., Lau, T.C., & Wu, R. Trace/heavy metal pollution monitoring in estuary and coastal area of Bay of Bengal, Bangladesh and implicated impacts. Mar. poll. bull. 2016, 105:393-402.

Khatun, N., Nayeem, J., deb, N., Hossain, S., & Kibria, M. M.  Heavy metals contamination: possible health risk assessment in highly consumed fish species and water of Karnafuli River Estuary, Bangladesh. Toxicology and Environmental Health Sciences, 2021, 13: 375-388.

Akila, M., Anbalagan, S., Lakshmisri, N. M., Janaki, V., Ramesh, T., Merlin, R. J., & Kamala-Kannan, S. Heavy metal accumulation in selected fish species from Pulicat Lake, India, and h Kibria, G,, Hossain, M,M,, Mallick, D., Lau, T.C., & Wu, R. Trace/heavy metal pollution monitoring in estuary and coastal area of Bay of Bengal, Bangladesh and implicated impacts. Mar. poll. bull. 2016, 105:393-402

Khatun, N., Nayeem, J., deb, N., Hossain, S., & Kibria, M. M.  Heavy metals contamination: possible health risk assessment in highly consumed fish species and water of Karnafuli River Estuary, Bangladesh. Toxicology and Environmental Health Sciences, 2021, 13: 375-388.

Sarker, M. J., Polash, A. U., Islam, M. A., Rima, N. N., & Farhana, T. (2020). Heavy metals concentration in native edible fish at upper Meghna River and its associated tributaries in Bangladesh: a prospective human health concern. SN Applied Sciences, 2, 1-13.

Ahmed, A. S., Rahman, M., Sultana, S., Babu, S. O. F., & Sarker, M. S. I. (2019). Bioaccumulation and heavy metal concentration in tissues of some commercial fishes from the Meghna River Estuary in Bangladesh and human health implications. Marine pollution bulletin, 145, 436-447.

Ali, M. M., Ali, M. L., Proshad, R., Islam, S., Rahman, Z., Tusher, T. R., ... & Al, M. A. (2020). Heavy metal concentrations in commercially valuable fishes with health hazard inference from Karnaphuli river, Bangladesh. Human and ecological risk assessment: an international journal, 26(10), 2646-2662.

Mohiuddin, M., Hossain, M. B., Ali, M. M., Hossain, M. K., Habib, A., Semme, S. A., … & Arai, T. (2022). Human health risk assessment for exposure to heavy metals in finfish and shellfish from a tropical estuary. Journal of King Saud University-Science, 34(4), 102035.ealth risk assessment. Environ. Technol. Innov. 2022, 27, 102744.

L83 " fish species (Otolithoides pama and Labeo bata)"Maybe you could make a short mention to its habitat (pelagic or benthis), feeding habits, etc...

Response: Thank you. We edited it as, “Both fish species are native to freshwater habitats, particularly rivers, streams, and floodplains as well as estuarine and coastal areas across Bangladesh.

Their feeding habits and other information are listed in the Table 1.

L85 "within the subtropical estuarine wetland system" Here I would write Feni river instead of "subtropical....system"? or a subtropical estuary with XXX characteristics? You may and should compare with other estuarine systems, but you will not be able to generalize the results for Feni River to any estuarine wetland system... 

Response: Thank you. We have written- Feni River estuary.

L86-87 "(ii) What health risks may arise from the consumption of these fish due to their toxic metal content"  Written as it is, it sounds speculative. You're not studying the effects on humans. You just calculating indexes. Better fuse with the previous question (i).

Response: Thank you very much. We fused it with (i).

The methods present serious flaws. My main concern is the experimental design and the interpretation of results, which do not support the conclusions. I believe that the number of individuals sampled per species and per site is too low. If the authors can consider each sampling place as replicates, then it will be acceptable if assumed to be a preliminary study or screenning. Otherwise, I don't think it can be published. Even if the number of samples individuals was higher the used stats are hardly enough to make the interpretation of these data. A PCA is adequate, but not made this way. 

Response:

Thank you for your concern. While we value your input, we are confident in the clarity and integrity of our methods and design. In this revised version, we have provided more detailed explanations of our methods and presented results to support our conclusions. Metal analysis methods are well-established, so we did not include exhaustive details.

For sample size estimation and species selection, we followed methods outlined in previous reputable publications, including our own papers (highly cited) in journals such as Toxics (Hossain et al., 2022), Chemosphere (Islam et al., 2018), etc. Our focus was on assessing metal levels in two commercially important fish species rather than habitat/site/species-specific variations. We considered all samples collectively, and the sample size of 36 was determined to be sufficient (Hossain et al., 2022) for reflecting contamination status, statistical evaluation, and health risk assessment. Our approach to health risk assessment involved integrating results from all samples. Therefore, we believe our methods, results, and conclusions are clear and informative.

For sources identification, we used the similar PCA analyses done by other previous studies.   

Hossain, M. B., Tanjin, F., Rahman, M. S., Yu, J., Akhter, S., Noman, M. A., & Sun, J. (2022). Metals bioaccumulation in 15 commonly consumed fishes from the lower Meghna river and adjacent areas of Bangladesh and associated human health hazards. Toxics, 10(3), 139.

In detail: L99 "the annual temperature" is that air temperature? What about water temperature?

Response: Thank you. Its air temperature. Data on annual air temperature are more readily available and easily accessible compared to detailed water temperature data. It might be more practical to use readily available data, especially if the study does not require precise water temperature measurements.

L102-105 Summarize

Response: Thank you. Summarized.

L108-112 Summarize

Response: Thank you. Summarized.

L133 "morpho characteristics" Delete

Response: The term "morpho characteristics" was deleted and written as follow:

“The weight and length of each collected fish were recorded, and subsequently, the edible portions and gills of each individual were stored separately.”

L137 "laboratory analysis procedures were detailed by Hossain et al." Neverthless you should summarize them here, with the necessary adaptations. 

Response: Thank you. We have added as below:

In the laboratory, each fish sample underwent cleaning and rinsing with deionized water. Subsequently, the fish were diced using a stainless-steel knife sanitized with acetone and hot distilled water prior to use. For metal analysis, both fish flesh and gills were placed in a beaker and subjected to ashing in a muffle furnace at 300°C for 3 hours. The resulting ash samples were then ground into a powder using a carbide mortar and pestle. The powdered sample was compressed into pellets measuring 2.5 cm in diameter using a hydraulic press pellet maker (Specac) with a pressure of 7 tons. Then the pellet was placed in EDXRF system for metal analysis. Irradiation of all samples was conducted according to a time-based program controlled by software provided with the EDXRF system. Standard materials underwent irradiation under identical experimental conditions to establish calibration curves for quantitative elemental determination in the respective samples.

L138 " in sediment samples" ???? I'm missing the description for sediment collection. Furthermore, there are no results on sediment samples in the Results section.

Response: Dear sir, we apologize for the typo mistake. We have corrected the sentence and rewritten as follow:

“The concentrations of various elements in fish samples were determined using energy dispersive X-ray fluorescence (EDXRF) spectrometry.”

L153 "The Target hazard quotient" There should be some more information about this index here or in the introduction

Response: Thank you.  we have mentioned some details in the introduction part.

L157-166 The explanation is not clear at all. Please re-write.

Response: Thank you, the explanations were rewritten as follow:

“EF represents exposure frequency (365 days/year) [30], ED denotes exposure duration (70 years for non-cancer risk, following USEPA guidelines [29] and Yi et al. [31]. FIR stands for fish ingestion rate (7 g/person/day) [32], CF is the conversion factor (0.2) used to convert fresh weight (Fw) to dry weight (Dw) [33]. CM represents heavy metal concentration in fish (mg/kg Dw), WAB denotes the average body weight (60 kg) [34], ATn represents the average exposure time for non-carcinogens (EF×ED) as utilized in characterizing non-cancer risk, and RfD is the reference dose of the metal (3.0×10−4 mg/kg/day for Hg, 1.0×10−3 mg/kg/day for Cd, 4.0×10−3 mg/kg/day for Pb, 3.0×10−4 mg/kg/day for As, and 4.0×10−2 mg/kg/day for Cu) [29]”

L161 " CM represents fish heavy metal concentration" For those who don't know the index this is not clear at all. To start with not saying the the quotient is calculated for each metal, not in the beggining nor in its parcels.

Response: Dear sir, we have corrected it.

L179 "10-4 to 10-6" -4 and -6 should be superscript

Response: Dear sir, thank you for such constructive comments. We have corrected the numbers.

L196  "identify probable origins of pollution" What do you mean by origin? Species? Place?

Response: Thank you. We meant sources.

Results and Discussion. In general, all the considerations about the possible sources of heavy metal fish contamination should be avoided in the Conclusions and Abstract sections as there is no data on the anthropogenic impact around the sampling sites and so they're not considered in the statistics. There is no discussion on the "differences" (not tested) between the two studied species. 

Response: Thank you for your valuable feedback. We acknowledge that only isotopic analyses can definitively confirm the origin of metals in the study area. However, previous studies conducted in the area or neighboring regions, as well as the land use pattern of the catchment area, can offer insights into potential sources of metals. In our study, we use the term 'potential or probable sources' to reflect this understanding. Additionally, similar methodologies to ours have been employed in numerous studies worldwide, as evidenced by publications such as Zhang et al., 2018, and other papers in reputable journals. We have also previously published a paper on the same river in Chemosphere, employing a similar approach. If further clarification is needed, we are open to making adjustments in our next revision.

For example,

Zhang, Z., Lu, Y., Li, H., Tu, Y., Liu, B., & Yang, Z. (2018). Assessment of heavy metal contamination, distribution and source identification in the sediments from the Zijiang River, China. Science of the Total Environment, 645, 235-243.

Islam, M. S., Hossain, M. B., Matin, A., & Sarker, M. S. I. (2018). Assessment of heavy metal pollution, distribution and source apportionment in the sediment from Feni River estuary, Bangladesh. Chemosphere, 202, 25-32.

Xie, F., Yu, M., Yuan, Q., Meng, Y., Qie, Y., Shang, Z., ... & Zhang, D. (2022). Spatial distribution, pollution assessment, and source identification of heavy metals in the Yellow River. Journal of Hazardous Materials, 436, 129309.

Fang, X., Peng, B., Wang, X., Song, Z., Zhou, D., Wang, Q., ... & Tan, C. (2019). Distribution, contamination and source identification of heavy metals in bed sediments from the lower reaches of the Xiangjiang River in Hunan province, China. Science of the Total Environment, 689, 557-570.

We have conducted ANOVA and edited it as, ‘the average metallic concentrations per species not varied significantly (p>0.05).’

L209 Labeo bata should be italic

Response: Made italic

L211-213 Re-write the whole sentence, please. It's very confusing.

Response: Thank you. Re-written

L214 Delete "However"

Response: Thank you. We have deleted.

L214-215 The differences between O.pama and L.bata should have been tested and discussed.

Response: Dear sir, thank you for such constructive comments. We have conducted ANOVA and edited it as, ‘the average metallic concentrations per species not varied significantly (p>0.05).’

Table 2 I can see the exact same values for Zn and Cu in muscle and gill. Please check this or explain.

Response: Thank you very much for noticing this. We have checked our main file again and found its almost same except for Zn value in muscle: 110. 85 ±1.236.

L221-222 There should be also a sentence with mention to O.pama and L.bata habitat and feeding habits.

Response: We have  added more information on it as per your suggestion.

L221-226 This discussion is relevant but it is cleary missing the comparison with the results herein presented with reference to  O.pama and L.bata habitat and feeding habits. You should be discussing this paper results, as compared to all those. The way it is written, it is just pointless. 

Response:  Thank you for your concern. We have included additional information on habitat and feeding habits in lines ; 315-320 and in Table 1. While we acknowledge your perspective to some extent, our main objectives centered on evaluating toxic metal concentrations in commonly consumed fish species and potential human health risks. Considering the scope and focus of our study, we did not incorporate an assessment of metal concentration in relation to the habitat and feeding habits of O. pama and L. bata. This decision was thoughtful, aiming to uphold clarity and focus within our research objectives. We may focus this aspect including some more species in our future studies.

L231 "in fish muscle" is this from all fish from both species and the 3 sites? It's not clear. Should be clarified and the approach should be explained.

Response: Thank you. Yes-  have modified it as, “While the concentrations of Zn and Cu in fish muscle were higher compared to the studied fish species from the Red Sea, Egypt [33], Mediterranean Sea, Turkey [48,49], and Gulf of Cambay, India [50], they remained within the standard values outlined by the FAO [51] indicating that despite higher levels in comparison to other regions, they did not exceed internationally recognized safety thresholds.”

L235-236 "were lower than those found in fish from the Bangshi River" You could discuss this comparison with Bangshi River a little more. Any data about this river? Compare the number of inhabitants, houses, activities in its water basin, with that Feni's basin?

Response: Thank you. We have modified it as, “In our study, the levels of Pb, Zn, and Cr observed in fish gills were notably lower than those reported in fish sampled from the contaminated Bangshi River [52]. This stark difference can be attributed to the pronounced contamination issues plaguing the Bangshi River, including unregulated discharge from the Dhaka Export Processing Zone (DEPZ) and the adjacency of pharmaceutical industries, poultry farms, and a tannery along its banks. In contrast, the Feni River estuary, while facing its own environmental challenges, does not exhibit the same degree of heavy metal pollution as observed in the Bangshi River.”

The number of inhabitants, houses, activities in its these water basins are not available in published paper/report.

L237 " on fish organs" which organs? Bibliographic references needed.

Response: Thank you. Fish muscle. We have added the references.

L238  "and serve as long-term sources of contamination..." Please re-write

Response: The sentence is rewritten as “Metals such as Pb and Hg are commonly associated with sediments and are vital sources of contamination for benthic fish”

L239-240 "Additionally, the presence of Hg in this study could be attributed to industrial discharges" That's a bit of a stretch. Your results do not support that conclusion. You can mention the possible Hg sources in Feni basin, without saying that the Hg levels found in this study are due to.... For example, the next sentence (on Cu and Cr) is just fine. 

Response:  Thank you very much. We  have modified  it as "Moreover the probable sources of Hg in the Feni River estuary are linked to industrial discharges and the deposition of atmospheric pollutants from coal-fired brickfields along the river [54].”

Table 3 Do these values refer to all fish together, 2 species and 3 sites? If so, it should be clearly stated on the table's legend. refers to the Standard values in the table?; because d never shows up in the table

Response: Thank your very much for pointing out this. Yes. We have modified the table legend. refers to the Standard values in the table. We have added ‘d’ in the table.

L247-253 Why not fuse this paragraph with 3.1?  The point of measuring heavy metal concentrations was assessing health risk coming from these species consumption, right?

Response:  Thank you. "Certainly, the main aim of measuring heavy metal concentrations in the studied species was to evaluate the potential health risks linked to their consumption. We maintained separate sections for this purpose. In section 3.1, we assessed the risk by comparing with other studies and standard values, while in section 3.2, we utilized established indices for the evaluation."

L247 "To assess the health risk to the local population from fish consumption, metal " Re-write please.

Response: Thank you. We have re-written as, “Metal concentrations in fish muscle were utilized to evaluate the potential health risks posed to the local population through fish consumption. Risk indices were computed by comparing these concentrations with the maximum permissible limits for human consumption set by the Food and Agriculture Organization [51].”

L263-264  "where an HI surpassing 1 raises concerns regarding health risks to the local populace"  The next sentence says the same, but more clearly.

Response: We have corrected it.

Table 4 Highlight HI, by putting it in bold, for example...

Response: We have highlighted.

L272 Re-write the whole sentence please.

Response: Re-written as ‘The Target Cancer Risk (TR) from consuming fish was assessed for Pb, Cr, and Zn. The average TR values for Pb, Cr, and Zn from the consumption of Otolithoides pama were’

L273 "and Zn from the consumption of Otolithoides pama" Re-write please

Response: Re-written as ‘The Target Cancer Risk (TR) from consuming fish was assessed for Pb, Cr, and Zn. The average TR values for Pb, Cr, and Zn from the consumption of Otolithoides pama were’

L278 Delete "However"

Response: Deleted.

L279 "while for Zn, it exceeded the limit" Re-write please.

Response: Thank you. Re-written as ‘However, for zinc (Zn), the risk exceeded the acceptable limit.’

L285 " EDI of heavy metals offers insight into the amount of these contaminants’ individuals are likely ingesting on a daily basis" Re-write please.

Response: We corrected it as, “The Estimated Daily Intake (EDI) of heavy metals provides valuable insight into the quantity of these contaminants that individuals are likely consuming on a daily basis.”

L286-288 "By emphasizing the levels of nutrients, contaminants and bioactive compounds consumed, this method provides significant understanding into possible dietary deficiencies or exposition to food allergens" Re-write please.

Response: We corrected it as, “This method highlights the levels of nutrients, contaminants, and bioactive compounds ingested, offering valuable insights into potential dietary deficiencies or exposure to food allergens [69].”

L294 "can potentially lead to health issues for consumers " Be cautios there. Your results do not support this conclusion. Just comparing with MTDI is enough. 

Response: Thank you for concern. We have re-written this whole section as below:

L295-296 "However, the quantity of fish consumed as well as the concentration of a particular metal in fish determines the quantity of that element can be identified in fish." Un-readable sentence.

Response: Thank you. We have re-written the whole section as below:

The Estimated Daily Intake (EDI) of heavy metals provides valuable insight into the quantity of these contaminants that individuals are likely consuming on a daily basis. This method highlights the levels of nutrients, contaminants, and bioactive compounds ingested, offering valuable insights into potential dietary deficiencies or exposure to food allergens [69].  The intake data can then be utilized to analyze a specific element of interest. This study assessed the dietary exposure to five trace elements through the consumption of fish in a regular human diet and measures the dietary intake. Table 6 displays the estimated daily intake (EDI) of heavy metals. Comparison with the maximum tolerable daily intake (MTDI) reveals that the levels of Pb, Hg, Zn, and Cr in the muscle tissue of the two fish species at almost all stations could potentially raise health concerns for consumers (See Table 6). Nonetheless, the severity of health issues depends on both the quantity of fish consumed and the concentration of a specific metal in the fish.

L304 "PCA can be beneficial in..." Re-write please.

Response: Thank you – re-written

L306 enough instead of "able"

Response: Corrected-the word "able " was replaced with “enough” as per your suggestion.

L309-311 " These results indicated that the presence of heavy metals in fish species was likely entering from various sources, which could be either anthropogenic (resulting from human activities) or natural" I don't understand how you can inferr this from your PCA analysis

Response: Thank you very much. We have re-written this section.

Generally, the PCA analysis likely revealed distinct clusters or patterns among the heavy metal concentrations in fish species. If the PCA results showed that the heavy metal concentrations varied across different fish species and did not cluster, based on specific characteristics such as habitat or diet, it suggests that the sources of heavy metals are diverse. Moreover, if certain heavy metals show high loadings on principal components associated with anthropogenic sources, it suggests a strong correlation between those metals and human activities.

L312-314"The wide-spread anthropogenic inputs from inundation croplands, pharmaceutical waste, household waste, upstream metal plating entities and airborne particles were visibly responsible for the high concentrations of Zn, Cu, and Hg" I don't see how you results support this conclusion..

Response: Thank you for your feedback. We have re-written the whole section.

Through PCA and Correlation analyses, we identified a significant correlation among Zn, Cu, and Hg. Consequently, we attempted to associate their origins with human activities within the catchment area, as discussed in Islam et al. (2018). Our findings revealed that the catchment area is predominantly utilized for agricultural activities, human habitation, fishing, metal plating, and pharmaceutical manufacturing, such as Globe Pharmaceuticals. Given the area's susceptibility to flooding and its location within a region characterized by heavy rainfall, the catchment areas become inundated during the rainy season, potentially transporting these metals to the estuary. If you look at the similar published paper, origination of metals is predicted in this way.

Table 7: Legend, "Shrimp"?; Are all the correlations significant? Not clear....

Response: We have corrected it. Zn-Cu-Hg correlations were significant at 5% level as indicated by *.

Conclusions section: some of the conclusions are not supported by the results, which one of these study's major flaws.

Response: We appreciate your perspective.  We have written the conclusion which clearly supported by results. We have distilled the essence of our actual findings to support our claim/objectives, ensuring that no extraneous or irrelevant information is included.

‘This study aimed to assess the contamination levels and associated health risks posed by five heavy metals (Pb, Hg, Zn, Cu, and Cr) in two commercially significant fish species from the Feni River estuary. Analysis of heavy metal concentrations in the muscle tissue of both fish species revealed a descending order of Zn > Cu > Hg > Pb > Cr. The highest concentration of Zn, Pb, and Cr were observed in the muscle tissue of Labeo bata, whereas Otolithoides pama exhibited the highest levels of Cu and Hg. The levels of studied metals in the gills of two species varied. Disparities in heavy metal presence in the fish species suggest that dietary habits and habitat may influence metal accumulation. While levels of Pb and Hg exceeded permissible thresholds for human consumption, assessments of Total Hazard Quotient (THQ), Hazard Index (HI), and Estimated Daily Intake (EDI) indicated that consuming fish from the studied region did not pose significant health risks, except for potential cancer hazards for Zn. The notable correlations among Zn, Cu, and Hg imply potential shared origins, whether anthropogenic or natural. Although isotopic analyses could provide definitive evidence of metal origins, possible sources may include wide-spread use of agrochemicals in croplands, household waste, upstream metal plating activities, and other human activities within the catchment area. Given that fish continue to be a vital and healthy component of a balanced diet, it is essential to recognize the potential human health risks of contaminated fish. Therefore, it is advisable to maintain regular monitoring of toxic metals in riverine fish. ’

L337 ", marking the first such study" Delete or re-phrase.

Response: We have corrected it- deleted.

L338 "that Labeo bata exhibited lower levels of heavy metal contamination compared to Otolithoides pama" Not clearly stated or discussed in the previous section.

Response: Re-written as ‘Analysis of heavy metal concentrations in the muscle tissue of both fish species revealed a descending order of Zn > Cu > Hg > Pb > Cr. The highest concentrations of Zn, Pb, and Cr were observed in the muscle tissue of Labeo bata, whereas Otolithoides pama exhibited the highest levels of Cu and Hg.’

L345 " risks apart from potential cancer risks" Re-write please.

Response: Thank you. We have re-written as below:

While levels of Pb and Hg surpassed the allowable threshold for human consumption, evaluations of Total Hazard Quotient (THQ), Hazard Index (HI), and Estimated Daily Intake (EDI) indicated that consuming fish from the examined region did not present substantial health risks, with the exception of potential cancer hazards.

L345-348 "Correlation and principal component analyses shed light on the potential sources of heavy metals in the environment, which included anthropogenic activities like the use of agricultural chemicals, silver nanoparticles, antimicrobial agents, and metallic plating industries along the estuary" Honestly, I don't see how this can inferred from the presented PCA analysis.

Response:  Thank you for insightful comments. We have specified the results from PCA and Correlation analyses and re-written as below:

The close correlations observed among the metals Zn, Cu, and Hg suggest that they may share common origins, stemming from either anthropogenic (human-induced) or natural sources. However, potential sources could include the extensive use of agrochemicals in croplands, household waste, metal plating activities upstream, and other human interventions within the catchment area.

L348-351 Re-write please.

Response: Thank you. Re-written as ‘Given that fish continue to be a vital and healthy component of a balanced diet, it's essential to recognize the potential human health risks. Therefore, it is advisable to maintain regular monitoring of toxic metals in riverine fish.’

Comments on the Quality of English Language

Overall the English is fine, except for some terms that in my opinion should be replaced. 

Response: Thank you for your feedback. We have improved the English language.

Reviewer 2 Report

Comments and Suggestions for Authors

The present MS is interesting and valuable for pollutant contamination monitoring in wild environments. 
Despite this, there are several major concerns that I would to see fixed before the possible publication.

Authors should increase the information provided in the introduction section. This is too short and misses some information regarding the studied species and the pollution history of the studied area. Moreover, the authors should add some information regarding other species used for pollutant monitoring.

Tables mus be imporved. In the Tab. 1 the standard deviation of the biometrics is missing.

Concerning the results and discussion section, I strongly suggest authors separate the two sections, to improve the clarity of the entire MS. Moreover, the Discussion should be enlarged. it is too short and superficial. 

I will provide a more detailed point-by-point comment list with the re-organized new manuscript.

Comments on the Quality of English Language

English language should be improved. 

Author Response

The present MS is interesting and valuable for pollutant contamination monitoring in wild environments.

Response: Thank you very much for your valuable time and comments.

Despite this, there are several major concerns that I would to see fixed before the possible publication.

Authors should increase the information provided in the introduction section. This is too short and misses some information regarding the studied species and the pollution history of the studied area. Moreover, the authors should add some information regarding other species used for pollutant monitoring.

Response: Thank you. We have re-written the introduction section as below:

We modified the introduction as below:  

Long-lasting toxic metals, recognized for their enduring presence in the environment, are significant pollutants capable of inducing detrimental effects such as cytotoxicity, mutagenicity, and carcinogenicity in organisms [1,2,3,4]. These contaminants penetrate aquatic ecosystems through various pathways, encompassing surface runoff, untreated wastewater and sewage discharge, deposition of airborne dust and aerosols, agricultural fertilizer application, electronic waste, and industrial effluents [5,6,7,8,9,10]. Ultimately, these metals enter into the food chain, accumulating in aquatic organisms and eventually in humans, leading to persistent adverse effects on human health [11,12].

Both humans and animals come into contact with toxic metals through various pathways of exposure, such as ingestion or dermal contact [11]. As an example, the toxicity of in-organic arsenic (As) can lead to issues such as abdominal pain, vomiting, and diarrhea. Lead (Pb), deemed a non-essential element, can have detrimental health impacts including liver and kidney damage, disruption of skeletal hematopoietic function, and ultimately, fatalities [12]. Chromium (Cr) plays a significant role in insulin function and lipid metabolism. Excessive intake of chromium may lead to pulmonary disorders along with liver and renal dysfunction [13]. Mercury (Hg) is deemed highly toxic, posing a lethal threat to both humans and other organisms. Ingesting high levels of zinc (Zn) can cause gastrointestinal symptoms such as nausea, vomiting, abdominal cramps, and diarrhea. Additionally, Prolonged exposure to elevated levels of zinc can potentially damage the liver and kidneys, leading to impaired liver function and kidney failure [11,14].

Fish, serving as a valuable source of high-quality protein and essential micronutrients, holds significant importance in the human diet, particularly in developing nations such as Bangladesh. Furthermore, fish are widely acknowledged as bio-indicators for assessing heavy metal contamination in aquatic environments. The metal concentrations found in fish tissues and organs reflect the levels of metals in water and their accumulation within the food chain [13]. Therefore, it is crucial to conduct investigations into the accumulation of potentially harmful heavy metals in key fish species and their various organs [14,15,16,17] to ensure that fish consumption does not become a pathway for the transfer of heavy metals to humans [18,19]. Otolithoides pama and Labeo bata are two widely distributed keystone species within estuarine habitats in Bangladesh, playing a vital role in ecosystem functioning and stability. As a commercially important fish species, these species support the livelihoods of many communities through fisheries and aquaculture activities, contributing to food security and economic development. Conservation efforts aimed at protecting of these species are essential for maintaining the ecological balance of estuarine ecosystems and sustaining the socio-economic well-being of dependent neighboring communities.

Regulatory bodies establish guidelines and standards for permissible levels of heavy metals in food, including fish. Assessing estimated daily intake (EDI), target hazard quotient (THQ), hazard index (HI) and cancer risk (CR) aids in evaluating compliance with these regulations. EDI estimation provides insight into the amount of heavy metals that individuals are likely to ingest daily through fish consumption. THQ, HI, and CR calculations further assess the potential health risks associated with this exposure, considering factors such as the concentration of contaminants in fish and consumption patterns. Exceeding allowable limits can result in regulatory actions and advisories to protect public health.

The Feni River estuary in Bangladesh plays an important role to accomplish for meeting local and national protein demands through by providing diversified fish and fishery products. However, the estuary is currently experiencing heavy metal contamination, with multiple contributing factors identified. These include population growth, agricultural practices, discharge of industrial and medical waste, haphazard population settlements, fish farming, washing activities, discharge of poultry waste, recreational pursuits, and improper disposal of untreated domestic effluents [20]. While numerous studies have investigated heavy metal contamination in fish globally [13, 14, 15, 16, 17, 18, 21], as well as in various estuaries in Bangladesh such as the lower Meghna estuary, upper Meghna estuary, and Karnaphuli estuary [4, 22, 23, 24, 25, 26, 27], there is a lack of data on heavy metal contamination and associated human health risks in edible fishes from the Feni River estuarine system. Previous research by Islam et al. [20] indicated sediment contamination in the estuary by certain metals and suggested further investigation at the organismal level. Thus, this study marks the initial step in assessing the concentrations of selected heavy metals (Pb, Hg, Zn, Cu, and Cr) in the muscle and gill tissues of two commonly consumed fish species (Otolithoides pama and Labeo bata) from the Feni River estuarine system. This study is important as it allows comparing Feni River with other aquatic systems, estuaries in particular and with other species, with similar niches. The study aims to explore the following inquiries: (i) what are the concentrations of toxic metals in commonly consumed fish species in the Feni river estuary and what health risks may arise from the consumption of these fish due to their toxic metal content? and (ii)What are the main sources of toxic metal contamination affecting these fish species?

Tables must be improved. In the Tab. 1 the standard deviation of the biometrics is missing.

Response: Thank you. SD values are added.

Concerning the results and discussion section, I strongly suggest authors separate the two sections, to improve the clarity of the entire MS. Moreover, the Discussion should be enlarged. it is too short and superficial.

Response: Thank you very much. We have added more explanation to interpret the results in this version. Please see track-changed version in section 3. Hope you will find it clearer and more intensive now.

We have combined the results and discussion following some other similar papers for several reasons:

 First of all, it helps readers understand the significance of the results within the larger research context by contextualizing the results by presenting them with their interpretation easily and clearly. Second, this method makes synthesis easier by allowing data and interpretation to be seamlessly integrated. This makes it easier to grasp how the results relate to answering research questions or validating hypotheses. Thirdly, it improves interpretation by allowing researchers to provide more in-depth analyses, justifications, and theories about patterns or trends they have noticed by directly tying results to their interpretation in the discussion section. Fourthly, it facilitates effective communication by simplifying the way results are presented, which makes it simpler for readers to understand the significance of the research findings. Finally, combined results and interpretation leads to a more comprehensive understanding of the study outcomes and their significance.

I will provide a more detailed point-by-point comment list with the re-organized new manuscript.

Response: Thank you. I will look forward to your comments. Although we belive this version will satisfy you.

Reviewer 3 Report

Comments and Suggestions for Authors

A brief summary

In this study authors assessed the bioaccumulation of Pb, Hg, Zn, Cu, and Cr in muscle and gills of Otolithoides pama and Labeo bata from the Feni estuary in Bangladesh. The obtained concentrations of invastigated metals in the muscle tissue were compared to the permissible limits for human consumption and a number of risk indices including estimated daily intake (EDI), target hazard quotient (THQ), hazard index (HI), and carcinogenic or target risk (TR), were calculated. The main findigs were: 1) the highest Zn concentrations in both muslce and liver of two fish species were recorded; 2) Pb and Hg were found to be above the permissible limits for human consumption; 3) the concentrations of Zn, Cu, and Cr in the muscle may pose threat according to EDI, as well as Zn according to TR; 4) there is no human health risk associated with the consumption of these species according to THQ and HI values; 5) Pb and Cr are of natural origin, while Zn, Cu, and Hg are from anthropogenic origin. The application of various indices when analyzing the human health risk is the main advantage of this manuscript.

The authors need to address the following comments before the paper get accept.

General comments

·         A statistical analysis of the obtained data should be done if the authors want to show which fish species is safer for human consumption and which species is better biondicator of metal pollution.

·         Authors did not indicate how and where sediment samples were collected (Materials and Methods section), nor did they present and discussed the concentrations of elements in the sediment (Results and Discussion section).

·         In Results and Discussion section, repetition of information from tables in the text should be avoided (i.e. concentration values).

·         In ″3.1. Heavy metals concentration″ focus on the significant differences between muscle and gills (if any) and try to explain them.

·         The paragraph 228-244 needs to be rewritten in ″3.1. Heavy metals concentration″. Compare your findings with other relevant studies dealing with fish species of similar ecology and commercial importance as O. pama and L. bata.

·         You provide full Latin name of the species when you mention it for the first time. After that, an abbreviated name is written which contains the first letter of the genus name and the full species name. For example, Otolithoides pama for the first time, and after that O. pama.

Specific comments

  • Lines 29-30: Explain how bioaccumulation of heavy metals in aquatic organisms (fish) may pose potential risks to human health.
  • Line 42: What is MTDI?
  • Lines 56-59: Indicate natural sources of heavy metals as well.
  • Lines 65-66: Add reference(s).
  • Lines 83-84: Provide information on the anual catchment of O. pama and L. bata. Also, provide information on ecology and feeding habits of these two species.
  • Lines 91-105: Add reference(s).
  • Lines 136-137: Give in more details.
  • Line 179: Replace 10-4 and 10-6 with 10-4 and 10-6.
  • Line 184: If any, provide CPSo values for analyzed metals.
  • Lines 184-185: Arsenic was not among the examined metals.
  • Lines 208-211: Is this statistically significant?
  • Line 209: Change ″Labeo bata″ to L. bata. Latin name of the species should be written in italic.
  • Line 211: Replace ″gill″ with gills.
  • Lines 211-214: Is this statistically significant?
  • Lines 228: It is more likely Asia-wide than worldwide.
  • Lines 273-275: Avoid the repetition. All values are already given in Table 5.
  • Lines 277-278: Replace 10-4 and 10-6 with 10-4 and 10-6.

Figures

  • The quality of Figure 1. needs to be improved.

Tables

  • Table 1:  SD values are missing.
  • Table 5: Replace ″E-″ with ″× 10-″.
  • Table 6: Provide unit for EDI.

Author Response

A brief summary

In this study authors assessed the bioaccumulation of Pb, Hg, Zn, Cu, and Cr in muscle and gills of Otolithoides pama and Labeo bata from the Feni estuary in Bangladesh. The obtained concentrations of invastigated metals in the muscle tissue were compared to the permissible limits for human consumption and a number of risk indices including estimated daily intake (EDI), target hazard quotient (THQ), hazard index (HI), and carcinogenic or target risk (TR), were calculated. The main findigs were: 1) the highest Zn concentrations in both muslce and liver of two fish species were recorded; 2) Pb and Hg were found to be above the permissible limits for human consumption; 3) the concentrations of Zn, Cu, and Cr in the muscle may pose threat according to EDI, as well as Zn according to TR; 4) there is no human health risk associated with the consumption of these species according to THQ and HI values; 5) Pb and Cr are of natural origin, while Zn, Cu, and Hg are from anthropogenic origin. The application of various indices when analyzing the human health risk is the main advantage of this manuscript.

Response: Thank you very much for your valuable time and insightful comments. We have carefully gone through all comments and addressed.

The authors need to address the following comments before the paper get accept.

General comments

  • A statistical analysis of the obtained data should be done if the authors want to show which fish species is safer for human consumption and which species is better biondicator of metal pollution.

Response: Thank you. We appreciate your constructive comments. Following your feedback, we conducted ANOVA analysis and determined that the average metallic concentrations per species did not vary significantly (p>0.05). This indicates that both species yielded similar results in terms of safety.

  • Authors did not indicate how and where sediment samples were collected (Materials and Methods section), nor did they present and discussed the concentrations of elements in the sediment (Results and Discussion section).

Response: Thank you. We apologize. We analyzed fish sample. In this study we assessed the accumulation of Pb, Hg, Zn, Cu, and Cr in two fish species from the Feni estuary in Bangladesh not sediment sanples.

  • In Results and Discussion section, repetition of information from tables in the text should be avoided (i.e. concentration values).

Response: Thank you. We have removed repetition following your instruction and remove repetition as much as required.

  • In ″3.1. Heavy metals concentration″ focus on the significant differences between muscle and gills (if any) and try to explain them.

Response:  Thank you. There were differences but no statistically significant variation was found between muscle and gills.

  • The paragraph 228-244 needs to be rewritten in ″3.1. Heavy metals concentration″. Compare your findings with other relevant studies dealing with fish species of similar ecology and commercial importance as O. pamaand L. bata.

Response: Thank you very much. We tried to compare with the studies from similar ecology and species. However, there are no such similar studies on the species.  

  • You provide full Latin name of the species when you mention it for the first time. After that, an abbreviated name is written which contains the first letter of the genus name and the full species name. For example, Otolithoides pama for the first time, and after that O. pama.

 Response: Thank you- we have modified as per your suggestions.

Specific comments

  • Lines 29-30: Explain how bioaccumulation of heavy metals in aquatic organisms (fish) may pose potential risks to human health.

Response: We have explained in lines : 87-97.

  • Line 42: What is MTDI?

Response: Maximum tolerable daily intake.

  • Lines 56-59: Indicate natural sources of heavy metals as well.

Response: Thank you indicated – geogenic.

  • Lines 65-66: Add reference(s).

Response: Added.

  • Lines 83-84: Provide information on the anual catchment of O. pama and L. bata. Also, provide information on ecology and feeding habits of these two species.

Response: Thank you.  Actually, we encountered limited information in the literature and from the Department of Fisheries regarding the annual catch of these specific species. Consequently, we have incorporated details about the ecology and feeding habits of these two species in both the introduction and discussion sections.

Lines 91-105: Add reference(s).

Response: Added

  • Lines 136-137: Give in more details.

Response: Thank you. We have added as below:

In the laboratory, each fish sample underwent cleaning and rinsing with deionized water. Subsequently, the fish were diced using a stainless-steel knife sanitized with acetone and hot distilled water prior to use. For metal analysis, both fish flesh and gills were placed in a beaker and subjected to ashing in a muffle furnace at 300°C for 3 hours. The resulting ash samples were then ground into a powder using a carbide mortar and pestle. The powdered sample was compressed into pellets measuring 2.5 cm in diameter using a hydraulic press pellet maker (Specac) with a pressure of 7 tons. Then the pellet was placed in EDXRF system for metal analysis. Irradiation of all samples was conducted according to a time-based program controlled by software provided with the EDXRF system. Standard materials underwent irradiation under identical experimental conditions to establish calibration curves for quantitative elemental determination in the respective samples.

  • Line 179: Replace 10-4 and 10-6 with 10-4 and 10-6.

Response: Dear sir, the numbers were replaced with the appropriate superscripts. Thank you.

  • Line 184: If any, provide CPSo values for analyzed metals.

Response: Thank you. Sorry- we do not have- its oral carcinogenic potency slope.

  • Lines 184-185: Arsenic was not among the examined metals.

Response: Yes – but it is used for the oral carcinogenic potency slope (CPSo) values for inorganic arsenic (As) in the calculation model. It is typically expressed as a numerical value representing the slope of the dose-response curve for carcinogenicity associated with oral exposure to inorganic arsenic.

  • Lines 208-211: Is this statistically significant?

Response: No.

  • Line 209: Change ″Labeo bata″ to L. bata. Latin name of the species should be written in italic.

Response: Dear sir, the name was corrected. Thank you

  • Line 211: Replace ″gill″ with gills.

Response: Dear sir, the term gill was replaced with gills as per your suggestion. Thank you.

  • Lines 211-214: Is this statistically significant?

Response: No

  • Lines 228: It is more likely Asia-wide than worldwide.

Response: Correct. We have corrected.

  • Lines 273-275: Avoid the repetition. All values are already given in Table 5.

Response: Thank you. Deleted from the text.

  • Lines 277-278: Replace 10-4 and 10-6 with 10-4 and 10-6.

 Response: The numbers were replaced with the appropriate superscripts. Thank you.

Figures

  • The quality of Figure 1. needs to be improved.

 Response: Figure 1 was replaced with high quality image, thank you.

Tables

  • Table 1:  SD values are missing.

Response: Thank you Added.

  • Table 5: Replace ″E-″ with ″× 10-″.

Response: Thank you. The letter “E” was replaced with “×10” as per your suggestion. thank you.

  • Table 6: Provide unit for EDI.

Response: Thank you added- mg/kg/day

Round 2

Reviewer 2 Report

Comments and Suggestions for Authors

The present MS has been improved according to the previous suggestions.

Despite this, there are still some major concerns regarding several parts of the paper.

Specific point-by-point comments have been provided below.

Line 125: please add to the scientific name the authorities. This is mandatory every first time in which a scientific name of a species is reported in a manuscript.

Results and Discussion section: I am still of the same idea. I think that results and discussion should be divided into two different sections, in this way it is confusing, at the expense, above all, of the discussions. But If authors prefer this MS organization, it is ok. Anyway, I strongly suggest authors improve the discussion of the results. It is still fragmentary and superficial, especially regarding the 3.2.2, 3.2.3 and 3.2.4 sections

Comments on the Quality of English Language

The English requires minor editing 

Author Response

The present MS has been improved according to the previous suggestions. Despite this, there are still some major concerns regarding several parts of the paper.

Response: Thank you again for your valuable time and insightful comments.

Specific point-by-point comments have been provided below.

Line 125: please add to the scientific name the authorities. This is mandatory every first time in which a scientific name of a species is reported in a manuscript.

Response: Thank you. We have added the authority :

Otolithoides pama (Hamilton, 1822)  and Labeo bata (Hamilton, 1822

Results and Discussion section: I am still of the same idea. I think that results and discussion should be divided into two different sections, in this way it is confusing, at the expense, above all, of the discussions. But If authors prefer this MS organization, it is ok. Anyway, I strongly suggest authors improve the discussion of the results. It is still fragmentary and superficial, especially regarding the 3.2.2, 3.2.3 and 3.2.4 sections.

Response: Thank you for your feedback. We acknowledge your perspective. As previously discussed in earlier revisions, we have chosen to combine the Results and Discussion sections. Our rationale for this decision stems from our belief that consolidating these sections enhances the clarity and comprehension of the research findings. By merging results and discussion, we aim to streamline the presentation of results, making it easier for readers to grasp the significance of our findings. We have previously published papers in ‘Biology’ following this approach, which we found effectively conveys the study outcomes and their implications ( for example; please see: Biology 202211(12), 1780; https://doi.org/10.3390/biology11121780)

We have significantly improved the discussion section ( 3.2.2, 3.2.3 and 3.2.4 sections)  as per your suggestions: please see the track-changed version. For example,

Section 3.2.2. : The Target Cancer Risk (TR) from consuming fish was assessed for Pb, Cr, and Zn. The TR for Pb, Zn, and Cr in fish is an important measure used to evaluate the potential health hazards of consuming fish contaminated with these heavy metals.  The metric indicates the projected likelihood of an individual acquiring cancer throughout their lifespan due to exposure to these particular pollutants. In this study, the average TR values for Pb, Cr, and Zn from the consumption of O. pama were found to be 2.632×10−6, 8.542×10−4 and 1.438×10−3, respectively (table 5). Conversely, in the case of L. bata, the TR values for Pb, Cr, and Zn were 3.422×10−6, 4.969×10−4, and 1.365×10−3, respectively (Table 5). These results indicate that the estimated carcinogenic risk associated with consuming L. bata is lower compared to O. pama  for the heavy metals analyzed. Although very low risks, it suggests that individuals consuming O. pama may have a slightly higher risk of developing cancer over their lifetime due to exposure to Cr, and Zn compared to those consuming L. bata.  As per established guidelines, a TR value of 1×10-6 is commonly used as a benchmark for acceptable risk in environmental and public health risk assessments. TR values below this threshold are typically considered to pose an acceptable level of risk and above cause significant risk. For fish, these TR values are typically grouped into three categories: TR < 10−6 is considered negligible, 10−6 < TR < 10−4 falls within an acceptable range, and TR > 10−4 is deemed unacceptable [36, 37, 68]. In this study, carcinogenic risk values for Pb and Cr were found to be within acceptable limits (table 5). However, for Zn, the risk exceeded the acceptable limit  (Table 5). Despite the examined fish species being considered safe for human consumption in the present study, there is a potential risk of developing cancer with continuous consumption over 70 years. Nevertheless, it is crucial to acknowledge that the definition of "safe" can differ based on the context and unique conditions of the exposure. Moreover, there may be variations in regulatory norms across different countries or areas. Hence, it is necessary to refer to pertinent standards and laws that are specific to the particular location of concern to ascertain safe TR values for fish intake.

Section 3.2.3. In order to evaluate the potential health risk to consumers, the findings were compared to the Maximum Tolerable Daily Intake (MTDI) for heavy metals (Table 6). The guideline value signifies the upper limit of a specific metal that an individual can ingest daily throughout their lifetime without encountering any negative health consequences [38]. It serves as a benchmark for evaluating the safety of consuming food and determining the acceptable quantities of heavy metals in food samples. Regulatory bodies normally determine MTDI values through scientific evaluations of the toxicity and health concerns linked to exposure to particular heavy metals.  In this study,  the EDI values for Hg and Zn in both fish species surpassed the maximum tolerable daily intake (MTDI) limit.  These findings suggests that people who eat these types of fish may be at risk of absorbing levels of mercury and zinc that beyond the recommended safe daily intake for a lifetime, which could lead to negative health consequences. The surpassing of the maximum tolerable daily intake levels for Hg and Pb raises concerns over the potential health hazards linked to the intake of fish, which may result in heavy metal exposure. Mercury (Hg) is specifically recognized for its ability to cause damage to the nervous system while consuming too much Zn can result in digestive problems and other health complications [37]. Hence, these findings emphasize the significance of monitoring the levels of heavy metals in fish and implementing steps to reduce exposure, in order to guarantee food safety and safeguard public health. Nevertheless, these figures consider variables such as an individual's body weight, the rate at which a substance is absorbed, and the possible cumulative consequences of prolonged exposure.

Section 3.2.4.

The PCA results show that the first component explained 69.79% of the total variation in the dataset, while the second component accounted for an additional 30.21% of the variance. Components with higher percentages of variance capture a greater amount of information regarding the underlying structure of the data. Hence, the first component, characterized by its significant variance percentage, accounts for a substantial proportion of the variability present in the dataset. The loadings of various metals on each component offer valuable insights into the interconnections between variables. In this instance, the first component displayed significant loadings of Zn and Cu, suggesting a robust link between these metals in the dataset. This indicates a strong correlation between the levels of Zn and Cu in the samples. In contrast, the second component showed a strong association with Hg, indicating that changes in Hg levels are less influenced by Zn and Cu levels and may indicate a unique pattern of variability in the dataset. In general, these findings indicate a strong correlation between the levels of Zn and Cu, although the variability in Hg contents differs among the analyzed samples.

Reviewer 3 Report

Comments and Suggestions for Authors

In Results and Discussion section, repetition of information from tables in the text should be avoided (i.e. concentration values).

Author Response

In Results and Discussion section, repetition of information from tables in the text should be avoided (i.e. concentration values)

Response: We greatly value your additional feedback. In this version, we have made intensive efforts to eliminate repetition as per your suggestions (specially deleting concentration values). We have streamlined the presentation by referencing only the most significant results extracted from the table in the text. This approach aims to simplify the results for general readers, enhancing their understanding of the key findings.